# Estimation of the firing behaviour of a complete motoneuron pool by combining electromyography signal decomposition and realistic motoneuron modelling

**Arnault H. Caillet** [1], **Andrew T. M. Phillips** [1], **Dario Farina**[2☯‡], **Luca Modenese** [1,3☯‡]*

**1** Department of Civil and Environmental Engineering, Imperial College London, United Kingdom,
**2** Department of Bioengineering, Imperial College London, United Kingdom, **3** Graduate School of
Biomedical Engineering, University of New South Wales, Sydney, Australia

☯ These authors contributed equally to this work.
‡ These authors are joint senior authors on this work.
* l.modenese@unsw.edu.au

UNITED STATES

**Data Availability Statement:** All relevant data
sources are specified within the manuscript. The
code used to generate the results is available at the

## Abstract

Our understanding of the firing behaviour of motoneuron (MN) pools during human voluntary muscle contractions is currently limited to electrophysiological findings from animal experiments extrapolated to humans, mathematical models of MN pools not validated for human data, and experimental results obtained from decomposition of electromyographical (EMG) signals. These approaches are limited in accuracy or provide information on only small partitions of the MN population. Here, we propose a method based on the combination of high-density EMG (HDEMG) data and realistic modelling for predicting the behaviour of entire pools of motoneurons in humans. The method builds on a physiologically realistic model of a MN pool which predicts, from the experimental spike trains of a smaller number of individual MNs identified from decomposed HDEMG signals, the unknown recruitment and firing activity of the remaining unidentified MNs in the complete MN pool. The MN pool model is described as a cohort of single-compartment leaky fire-and-integrate (LIF) models of MNs scaled by a physiologically realistic distribution of MN electrophysiological properties and driven by a spinal synaptic input, both derived from decomposed HDEMG data. The MN spike trains and effective neural drive to muscle, predicted with this method, have been successfully validated experimentally. A representative application of the method in MN-driven neuromuscular modelling is also presented. The proposed approach provides a validated tool for neuroscientists, experimentalists, and modelers to infer the firing activity of MNs that cannot be observed experimentally, investigate the neuromechanics of human MN pools, support future experimental investigations, and advance neuromuscular modelling for investigating the neural strategies controlling human voluntary contractions.

following public GitHub repository: https://github.com/ArnaultCAILLET/Caillet-et-al-2022-PLOS_Comput_Biol.

**Funding:** Skempton Scholarship to AC; Imperial College Research Fellowship to LM. LM also acknowledges support of the UNSW Scientia Fellowship scheme. The funders had no role in study design, data collection and analysis, decision to publish, or preparation of the manuscript.

**Competing interests:** The authors have declared that no competing interests exist.

## Author summary

Our experimental understanding of the firing behaviour of motoneuron (MN) pools during human voluntary muscle contractions is currently limited to the observation of small samples of active MNs obtained from EMG decomposition. EMG decomposition therefore provides an important but incomplete description of the role of individual MNs in the firing activity of the complete MN pool, which limits our understanding of the neural strategies of the whole MN pool and of how the firing activity of each MN contributes to the neural drive to muscle. Here, we combine decomposed high-density EMG (HDEMG) data and a physiologically realistic model of MN population to predict the unknown recruitment and firing activity of the remaining unidentified MNs in the complete MN pool. In brief, an experimental estimation of the synaptic current is input to a cohort of MN models, which are calibrated using the available decomposed HDEMG data, and predict the MN spike trains fired by the entire MN population. This novel approach is experimentally validated and applied to muscle force prediction from neuromuscular modelling.

## Introduction

During voluntary muscle contractions, pools of spinal alpha-motoneurons (MNs) convert the synaptic input they receive into a neural command that drives the contractile activity of the innervated muscle fibres, determining limb motion. Identifying the recruitment and firing dynamics of MNs is fundamental for understanding the neural strategies controlling human voluntary motion, with applications in sport sciences [1–3], and neurological and musculo-skeletal rehabilitation [4–8]. Determining the MN-specific contributions to the MN population activity also allows more realistic control of neuromuscular models [9–12], investigation of muscle neuromechanics [13,14], prediction of limb motion from MN-specific behaviour [15], or improvement in human-machine interfacing and neuroprosthetics [16,17].

Our understanding of MN pool firing behaviour during human voluntary tasks is however currently limited. While the MN membrane afterhyperpolarization and axonal conduction velocity can be inferred from indirect specialized techniques [18,19], most of the other electro-chemical MN membrane properties and mechanisms that define the MN recruitment and discharge behaviour cannot be directly observed in humans in vivo. Analysis of commonly adopted bipolar surface EMG recordings, which often lump the motor unit (MU) trains of action potentials into a single signal assimilated as the neural drive to muscle, cannot advance our understanding of the MN pool activity at the level of single MNs. Our experimental knowledge on the remaining MN membrane properties in mammals is therefore obtained from in vitro and in situ experiments on animals [20]. The scalability of these mechanisms to humans is debated [21] due to a systematic inter-species variance in the MN electrophysiological properties in mammals [22]. Decomposition of high-density EMG (HDEMG) or intramuscular EMG (iEMG) signals [23,24] allows the in vivo decoding in human muscles of the firing activity of individual active motoneurons during voluntary contractions and provide a direct window on the internal dynamics of MN pools. Specifically, the non-invasive EMG approach to MN decoding has recently advanced our physiological understanding of the neurophysiology of human MU pools and of the interplay between the central nervous system and the muscle contractile machinery [25,26]

Yet, the activity of all the MNs constituting the complete innervating MN population of a muscle cannot be identified with this technique. High-yield decomposition typically detects at

most 30-40 MNs [27], while MU pools typically contain hundreds of MUs in muscles of the hindlimb [20]. The small sample of recorded MNs is besides not representative of the continuous distribution of the MN electrophysiological properties in the complete MN pool with a bias towards the identification of mainly high-threshold MUs. The samples of spike trains obtained from signal decomposition therefore provide a limited description of the role of individual MNs in the firing activity of the complete MN pool.

To allow the investigation and description of specific neurophysiological mechanisms of the complete MN population, some studies have developed mathematical frameworks and computational models of pools of individual MNs. These MN pool models have provided relevant insights for interpreting experimental data [28,29], investigating the MN pool properties and neuromechanics [30–32], neuromuscular mechanisms [12,33], and the interplay between muscle machinery and spinal inputs [34]. However, none of these MN pool models have been tested with experimental input data, instead either receiving artificial gaussian noise [32], sinusoidal [31] or ramp [29] inputs, inputs from interneurons [30] or feedback systems [28]. These MN pool models have therefore never been tested in real conditions of voluntary muscle contraction. The forward predictions of MN spike trains or neural drive to muscle obtained in these studies were consequently not or indirectly validated against experimental recordings.

The MN-specific recruitment and firing dynamics of these MN pool models are usually described with comprehensive or phenomenological models of MNs. The biophysical approaches [30–33], which rely on a population of compartmental Hodgkin-Huxley-type MN models provide a comprehensive description of the microscopic MN-specific membrane mechanisms of the MN pool and can capture complex nonlinear MN dynamics [35]. However, these models are computationally expensive and remain generic, involving numerous electrophysiological channel-related parameters for which adequate values are difficult to obtain in mammalian experiments [36,37] and must be indirectly calibrated or extrapolated from animal measurements in human models [38]. On the other hand, phenomenological models of MNs [10,28,39] provide a simpler description of the MN pool dynamics and rely on a few parameters that can be calibrated or inferred in mammals including humans. They are inspired from the Fuglevand's formalism [29], where the output MN firing frequency is the gaussian-randomized linear response to the synaptic drive with a MN-specific gain. However, these phenomenological models cannot account for the MN-specific nonlinear mechanisms that dominate the MN pool behaviour [20,35,40,41]. MN leaky integrate-and-fire (LIF) models, the parameters of which can be defined by MN membrane electrophysiological properties for which mathematical relationships are available [42], are an acceptable trade-off between Hodgkin-Huxley-type and Fuglevand-type MN models with intermediate computational cost and complexity, and accurate descriptions of the MN macroscopic discharge behaviour [43] without detailing the MN's underlying neurophysiology [44]. While repeatedly used for the modelling and the investigation of individual MN neural dynamics [45–47], MN LIF models are however not commonly used for the description of MN pools.

To the authors' knowledge there is no systematic method to record the firing activity of all the MNs in a MN pool, or to estimate from a sample of experimental MN spike trains obtained from signal decomposition the firing behaviour of MNs that are not recorded in the MN pool. There is no mathematical model of a MN pool that (1) was tested with experimental neural inputs and investigated the neuromechanics of voluntary human muscle contraction, (2) involves a cohort of MN models that relies on MN-specific profiles of inter-related MN electrophysiological properties, (3) is described by a physiologically-realistic distribution of MN properties that is consistent with available experimental data. This limits our understanding of the neural strategies of the whole MN pool and of how the firing activity of each MN contributes to the neural drive to muscle.

In this study, a novel four-step approach is designed to predict, from the neural information of $N_r$ MN spike trains obtained from HDEMG signal decomposition, the recruitment and firing dynamics of the $N-N_r$ MNs that were not identified experimentally in the investigated pool of $N$ MNs. The model of the MN pool was built upon a cohort of $N$ single-compartment LIF models of MNs. The LIF parameters are derived from the available HDEMG data, are MN-specific, and account for the inter-relations existing between mammalian MN properties [42]. The distribution of $N$ MN input resistances hence obtained defines the recruitment dynamics of the MN pool. The MN pool model is driven by a common synaptic current, which is estimated from the available experimental data as the filtered cumulative summation of the $N_r$ identified spike trains. The MN-specific LIF models phenomenologically transform this synaptic current into accurate discharge patterns for the $N_r$ MNs experimentally identified and predict the MN firing dynamics of the $N-N_r$ unidentified MNs. The blind predictions of the spike trains of the $N_r$ identified MNs and the effective neural drive to the muscle, computed from the firing activity of the complete pool of $N$ virtual MNs, are both successfully validated against available experimental data.

Neuroscientists can benefit from this proposed approach for inferring the neural activity of MNs that cannot be observed experimentally and for investigating the neuromechanics of MN populations. Experimenters can use this approach for better understanding their experimental dataset of $N_r$ identified discharging MNs. Moreover, this approach can be used by modelers to design and control realistic neuromuscular models, useful for investigating the neural strategies in muscle voluntary contractions. In this study, we provide an example for this application by using the simulated discharge patterns of the complete MN pool as inputs to Hill-type models of muscle units to predict muscle force.

## Methods

### Overall approach

The spike trains $sp_j^{sim}(t)$ elicited by the entire pool of $N$ MNs were inferred from a sample of $N_r$ experimentally identified spike trains $sp_i^{exp}(t)$ with a 4-step approach displayed in Fig 1. The $N_r$ experimentally-identified MNs were allocated to the entire MN pool according to their recorded force recruitment thresholds $F_i^{th}$ (step 1). The common synaptic input to the MN pool was estimated from the experimental spike trains $sp_i^{exp}(t)$ of the $N_r$ MNs (step 2) and was linearly scaled for simplicity to the total postsynaptic membrane current $I(t)$ responsible for spike generation. A cohort of $N_r$ single-compartment leaky-and-fire (LIF) models of MNs, the electrophysiological parameters of which were mathematically determined by the unique MN size parameter $S_i$, transformed the input $I(t)$ to simulate the experimental spike trains $sp_i^{exp}(t)$ after calibration of the $S_i$ parameter (step 3). The distribution of the $N$ MN sizes $S(j)$ in the entire MN pool, which was extrapolated by regression from the $N_r$ calibrated $S_i$ quantities, scaled the electrophysiological parameters of a cohort of $N$ LIF models. The $N$ calibrated LIF models predicted from $I(t)$ the spike trains $sp_j^{sim}(t)$ of action potentials elicited by the entire pool of $N$ virtual MNs.

The following assumptions were made. (1) The MU pool is idealized as a collection of $N$ independent MUs that receive a common synaptic input and possibly MU-specific independent noise. (2) In a pool of $N$ MUs, $N$ MNs innervate $N$ muscle units (mUs). (3) In our notation, the pool of $N$ MUs is ranked from $j = 1$ to $j = N$, with increasing recruitment threshold. For the $N$ MNs to be recruited in increasing MN and mU size and recruitment thresholds according to Henneman's size principle [20,48–53], the distribution of morphometric,

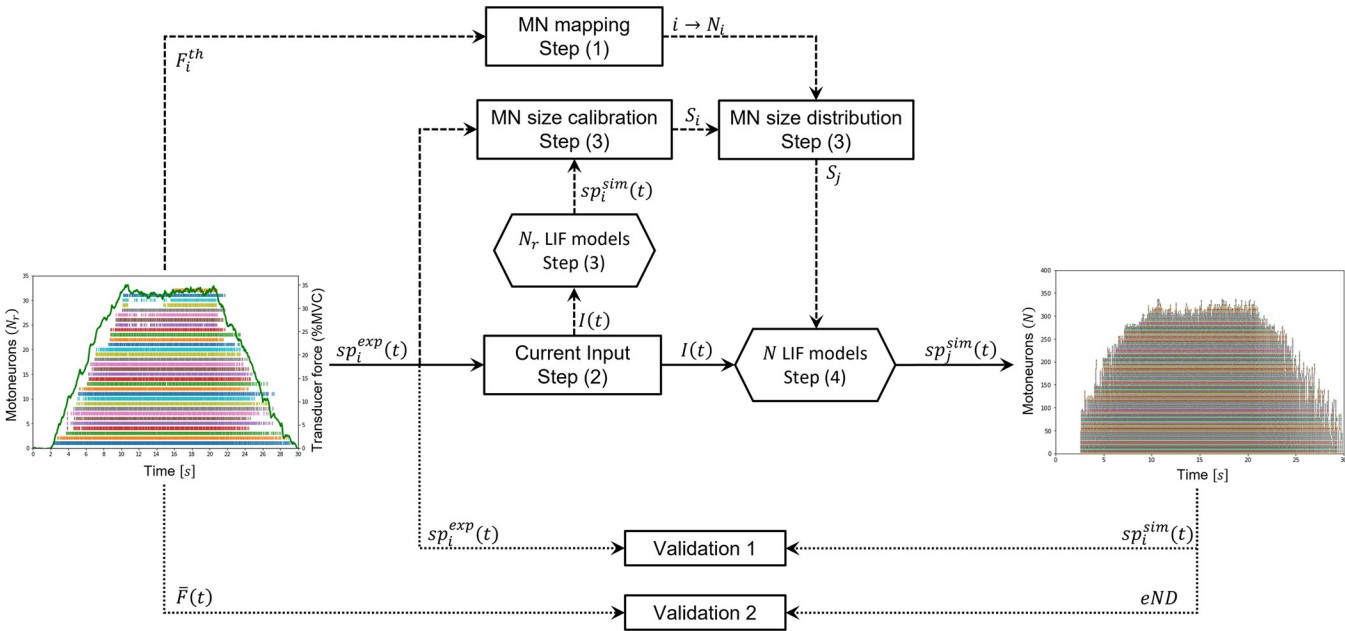

**Fig 1.** Four-step workflow predicting the spike trains $sp_j^{sim}(t)$ of the entire pool of $N$ MNs (right figure) from the experimental sample of $N_r$ MN spike trains $sp_i^{exp}(t)$ (left figure). Step (1): according to their experimental force thresholds $F_i^{th}$, each MN, ranked from $i = 1$ to $i = N_r$ following increasing recruitment thresholds, was assigned the $N_i^{th}$ location in the complete pool of MNs ($i \rightarrow N_i$ mapping). Step (2): the current input $I(t)$ common to the MN pool was derived from the $N_r$ spike trains $sp_i^{exp}(t)$. Step (3): using $I(t)$ as input, the size parameter $S_i$ of a cohort of $N_r$ leaky-and-fire (LIF) MN models was calibrated by minimizing the error between predicted and experimental filtered spike trains. From the calibrated $S_i$ and the MN $i \rightarrow N_i$ mapping, the distribution of MN sizes $S_j$ in the entire pool of virtual MNs was obtained by regression. Step (4): the $S_j$ distribution scaled a cohort of $N$ LIF models which predicted the MN-specific spike trains $sp_j^{sim}(t)$ of the entire pool of MNs (right). The approach was validated by comparing experimental and predicted spike trains (Validation 1) and by comparing normalized experimental force trace $\overline{F}(t)$ (Left figure, green trace) with normalized effective neural drive (Validation 2). In both figures, the MN spike trains are ordered from bottom to top in the order of increasing force recruitment thresholds.

threshold and force properties in the MN pool follows

$$\forall j, k \in [1; N], (j < k) \Longrightarrow (S_j < S_k \Longleftrightarrow I_j^{th} < I_k^{th} \Longleftrightarrow IR_j < IR_k \Longleftrightarrow F_j^{th} < F_k^{th} \Longleftrightarrow f_{iso,j}^{max} < f_{iso,k}^{max})$$

where $S$ is the MN surface area, $I^{th}$ the MN current threshold for recruitment, $IR$ the MU innervation ratio defining the MU size, $F^{th}$ is the MU force recruitment threshold, and $f_{iso}^{max}$ is the MU maximum isometric force. (4) The MN-specific electrophysiological properties are mathematically defined by the MN size $S$ [42]. This extends the Henneman's size principle to:

$$\forall j, k \in [1; N], (j < k) \Longrightarrow (S_j < S_k \Longleftrightarrow C_j < C_k \Longleftrightarrow R_j > R_k \Longleftrightarrow \tau_j > \tau_k)$$

Where $C$ is the MN membrane capacitance, $R$ the MN input resistance and $\tau$ the MN membrane time constant.

## Experimental data

The four sets of experimental data used in this study, named as reported in the first column of Table 1, provide the time-histories of recorded MN spike trains $sp_i^{exp}(t)$ and whole muscle force trace $F(t)$ (left panel in Fig 1), and were obtained from the studies [26,27,54,55], as open-source supplementary material and personal communication, respectively. In these studies, the HDEMG signals were recorded with a sampling rate $f_s = 2048 Hz$ from the Tibialis Anterior (TA) and Gastrocnemius Medialis (GM) human muscles during trapezoidal isometric contractions. As displayed in Fig 2, the trapezoidal force trajectories are described in this study by

**Table 1. The four experimental datasets processed in this study.** $N_r$ spike trains are identified per dataset during trapezoidal contractions of the Tibialis Anterior (TA) or Gastrocnemius Medialis (GM) muscles. The trapezoidal force trace is described by times $t_{tr_i}$ in seconds up to a dataset-specific level of maximum voluntary contraction (%MVC).

| Dataset | Muscle | %MVC | $t_{tr0}$ | $t_{tr1}$ | $t_{tr2}$ | $t_{tr3}$ | $t_{tr4}$ | $t_{tr5}$ | $N_r$ | Reference paper |
|---------|--------|------|-----------|-----------|-----------|-----------|-----------|-----------|-------|-----------------|
| $DTA_{35}$ | TA | 35 | 0 | 2.2 | 10.6 | 20.5 | 30 | 30 | 32 | [26,27] |
| $HTA_{35}$ | TA | 35 | 0 | 2.1 | 10.5 | 20.5 | 30 | 33 | 21 | [54,55] |
| $HTA_{50}$ | TA | 50 | 0 | 1.6 | 12 | 21.8 | 34.5 | 35 | 14 | |
| $HGM_{30}$ | GM | 30 | 0 | 3.1 | 9.1 | 28 | 33.5 | 107 | 27 | |

the $t_{tr_{0 \to 5}}$ times reported in Table 1, as a zero force in $[t_{tr0}; t_{tr1}]$, a ramp of linearly increasing force in $[t_{tr1}; t_{tr2}]$, a plateau of constant force in $[t_{tr2}; t_{tr3}]$, a ramp of linearly decreasing force in $[t_{tr3}; t_{tr4}]$, and a zero force in $[t_{tr4}; t_{tr5}]$.

The HDEMG signals were decomposed with blind-source separation techniques and $N_r$ MN spike trains $sp_i^{exp}(t)$ were identified. In this study, the experimental $N_r$ MNs were ranked from $i = 1$ to $i = N_r$ in the order of increasing recorded force recruitment thresholds $F_i^{th}$, i.e. $\forall i \in [1; N_r]$, $F_i^{th} < F_{i+1}^{th}$. The sample time of the $k^{th}$ firing event of the $i^{th}$ identified MN is noted as $ft_i^k$, and the binary spike train of the $i^{th}$ identified MN was mathematically defined as:

$$\begin{cases} sp_i^{exp}(t = ft_i^k) = 1 \\ sp_i^{exp}(t \neq ft_i^k) = 0 \end{cases} \quad \text{(Eq 1)}$$

The train of instantaneous discharge frequency $IDF_i(t)$ of the $i^{th}$ identified MN was computed between firing times $ft_i^k$ and $ft_i^{k+1}$ as:

$$\begin{cases} IDF_i(t = ft_i^k) = \dfrac{1}{ft_i^{k+1} - ft_i^k} \\ IDF_i(t \neq ft_i) = 0 \end{cases} \quad \text{(Eq 2)}$$

The IDFs were moving-average filtered by convolution with a Hanning window of length 400ms [56], yielding the continuous filtered instantaneous discharge frequencies (FIDFs) for all $N_r$ identified MNs.

## Approximation of the TA and GM MU pool size

The typical number $N$ of MUs was estimated for the TA muscle from cadaveric studies [57], statistical methods [58], decomposed-enhanced spike-triggered-averaging (DESTA) approaches [59–64], and adapted multiple point stimulation methods [65] in 20-80-year-old human subjects. Because of method-specific limitations [66], results across methods varied substantially, with estimates for $N$ of, respectively, 445, 194, 190, 188 and 300 MUs for the TA muscle. DESTA methods systematically underestimate the innervation ratio due to the limited muscle volume covered by the surface electrodes. Cadaveric approaches rely on samples of small size and arbitrarily distinguish alpha from gamma MNs. Twitch torque measurements are an indirect method for estimating $N$. Accounting for these limitations, we estimated the potentially conservative $N_{TA} = 400$ MUs in a typical adult TA muscle. Assuming 200,000 fibres in the TA muscle [67,68], $N_{TA} = 400$ yields a mean of 500 fibres per TA MU, consistently with previous findings [68]. In two cadaveric studies [57,69], the estimate for the GM was $N_{GM} = 550$ MUs, which is consistent with $N_{TA} = 400$ as the GM muscle volume is typically larger than TA's [70]. A sensitivity analysis on the performance of the method presented in Fig 1 upon variations on the value of $N_{TA}$ was included in this study.

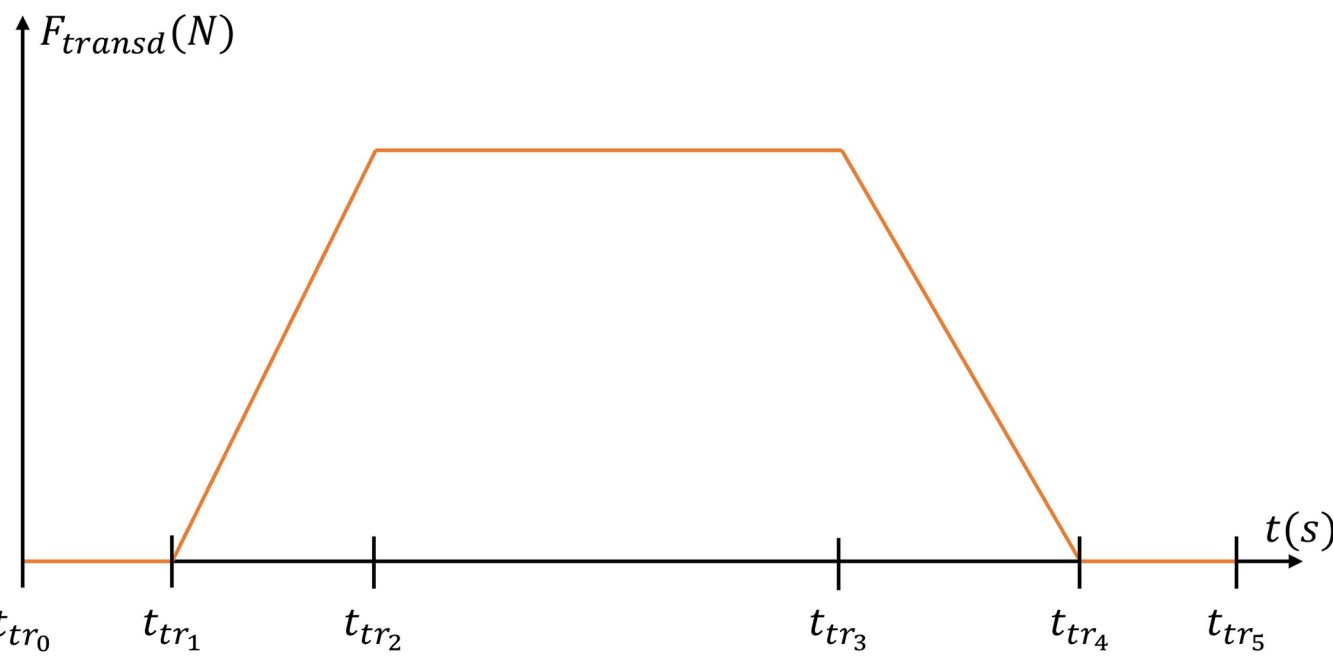

**Fig 2. Definition of times $t_{tr_i}$ that describe the trapezoidal shape of the muscle isometric contraction.**

## Step (1): MN mapping

In the first step of the approach overviewed in Fig 1, the $N_r$ experimentally identified MNs were allocated to the entire pool of $N$ MNs according to their recorded force recruitment thresholds $F_i^{th}$. Three studies measured in the human TA muscle *in vivo* the force recruitment thresholds $F^{th}$ of MUs, given as a percentage of the maximum voluntary contraction (%MVC) force, for 528 [64], 256 [71], and 302 [72] MUs. Other studies investigated TA MU pools but reported small population sizes [73] and/or did not report the recruitment thresholds [74–76].

We digitized the scatter plot in Fig 3 in [64] using the online tool WebPlotDigitizer [77]. The normalized MU population was partitioned into 10%-ranges of the values of $F^{th}$ (in % MVC), as reported in [71,72]. The distributions obtained from these three studies were averaged. The normalized frequency distribution by 10%MVC-ranges of the $F^{th}$ quantities hence obtained was mapped to a pool of $N$ MUs, providing a step function relating each $j^{th}$ MU in the MU population to its 10%-range in $F^{th}$. This step function was least-squares fitted by a linear-exponential trendline (Eq 3) providing a continuous frequency distribution of TA MU recruitment thresholds in a MU pool that reproduces the available literature data.

$$F^{th}(j) = k_1 \cdot \left( k_2 \cdot \frac{j}{N} + \Delta_F^{\left(\frac{j}{N}\right)^{k_3}} \right), j \in [1; N] \qquad \text{(Eq 3)}$$

Simpler trendlines, such as $F^{th}(j) = k_1 \cdot \Delta_F^{\left(\frac{j}{N}\right)^{k_3}}$ [29], returned fits of lower $r^2$ values. According to the three studies and to [20], a $\Delta_F = 120$-fold range in $F^{th}$ was set for the TA muscle, yielding $F^{th}(N) = 90\%\text{MVC} = \Delta_F \cdot F^{th}(1)$, with $F^{th}(1) = 0.75\%\text{MVC}$. Finally, the equation $F^{th}(N_i) = k_1 \cdot \left( k_2 \cdot \frac{N_i}{N} + \Delta_F^{\left(\frac{N_i}{N}\right)^{k_3}} \right) = F_i^{th}$ was solved for the variable $N_i$ for all $N_r$ identified MUs for which the experimental threshold $F_i^{th}$ was recorded. The $N_r$ identified MUs were thus

assigned the $N_i^{th}$ locations of the complete pool of $N$ MUs ranked in order of increasing $F^{th}$:

$$i \in [1; N_r] \rightarrow N_i \in [1; N]$$

When considering a 100-ms electromechanical delay between MN recruitment time and onset of muscle unit force, the mapping $i \in [1; N_r] \rightarrow N_i \in [1; N]$ did not substantially change. It was therefore simplified that a MN and its innervated muscle unit were recruited at the same time, and the $i \in [1; N_r] \rightarrow N_i \in [1; N]$ mapping derived for muscle units was extrapolated to MNs.

Considering that typically less than 30 MUs (5% of the GM MU pool) can be currently identified by HDEMG decomposition in GM muscles [27], and that the few papers identifying GM MUs with intramuscular electrodes either did not report the MU $F^{th}$ [78–80] or identified less than 24 MUs up to 100% MVC [81,82], a GM-specific $F^{th}(j)$ distribution could not be obtained from the literature for the GM muscle. The $F^{th}(j)$ distribution obtained for the TA muscle was therefore used for the simulations performed with the GM muscle, which is acceptable as an initial approximation based on visual comparison to the scattered data provided in these studies.

## Step (2): Current input $I(t)$

In the second step of the approach in Fig 1, the common synaptic input to the MN pool was first estimated from the experimental data. The cumulative spike train (CST) was obtained as the temporal binary summation of the $N_r$ experimental spike trains $sp_i^{exp}(t)$.

$$CST = \sum_i sp_i^{exp}(t) \qquad (Eq\ 4)$$

The effective neural drive (eND) to the muscle was estimated by low-pass filtering the CST in the bandwidth $[0; 10]Hz$ relevant for force generation [47]. As the pool of MNs filters the MN-specific independent synaptic noise received by the individual MNs and linearly transmits the common synaptic input (CSI) to the MN pool [31,83], the CSI was equalled to the eND in arbitrary units:

$$CSI(t) = k \cdot eND(t) \qquad (Eq\ 5)$$

The common synaptic control (CSC) signal was obtained by low pass filtering the CSI in $[0; 4]Hz$.

This approach, which estimates the CSI from the $N_r$ experimental spike trains is only valid if the sample of $N_r$ MNs is 'large enough' and 'representative enough' of the complete MN pool for the linearity properties of the population of $N_r$ MNs to apply [83]. To assess if this approximation holds with the $N_r$ MNs obtained experimentally, the following two validations were performed. (1) The coherence $coher_{\frac{N_r}{2}} \in [0; 1]$, averaged in $[1; 10]Hz$, was calculated between two cumulative spike trains $CST_{\frac{N_r}{2},1}$ and $CST_{\frac{N_r}{2},2}$ computed from two complementary random subsets of $\frac{N_r}{2}$ MNs. This was repeated 20 times for random permutations of complementary subsets of $\frac{N_r}{2}$ MNs, and the $coher_{\frac{N_r}{2}}$ values were average yielding $coher_{\frac{N_r}{2},mean} \in [0; 1]$. The coherence $coher_{N_r}$ between the complete experimental sample of $N_r$ MNs and a virtual sample of $N_r$ non-identified MNs was finally estimated by reporting the pair $\left(\frac{N_r}{2}; coher_{\frac{N_r}{2},mean}\right)$ similarly to Fig 2A in [47]. (2) The time-histories of the normalized force $\overline{F}(t)$ and common synaptic control $\overline{CSC}$, which should superimpose if the linearity properties apply (see Fig 6 in [83]), were compared with calculation of the normalized root-mean-square error (nRMSE) and coefficient of determination $r^2$. If $coher_{N_r} > 0.7$, $r^2 > 0.7$ and $nRMSE < 30\%$, it was assumed that the sample

of $N_r$ MNs was large and representative enough of the MN pool for the linearity properties to apply, and the *eND* was confidently assimilated as the *CSI* to the MN pool in arbitrary units. It must be noted that if $coher_{N_r} < 1$, the linearity properties do not fully apply for the sample of $N_r$ MNs, and the *CSI* computed from the $N_r$ MNs is expected to relate to the true *CSI* with a coherence close but less than $coher_{N_r}$.

The *CSI* hence obtained reflects the net excitatory synaptic influx common to the MN population, which triggers the opening of voltage-dependent dendritic channels and the activation of intrinsic membrane currents, such as persistent inward currents (PICs), which contribute to the total dendritic membrane current $I(t)$ responsible for spike generation. As the single-compartment LIF model considered in this study does not describe the dendritic activity, the PIC-related nonlinearity in the *CSI–I* transformation was excluded from the model and $I(t)$ was simplified to be linearly related to *CSI* with a constant gain $G$ across the MN pool. To avoid confusion, $I(t)$ is referred as 'current input' in the following. The limitations of this simplification are investigated in the Limitations Section (Limitation 3). To identify $G$, it must be noted that the *CSI*, which was computed from a subset of the MN pool, does not capture the firing activity of the MNs that are recruited before the smallest identified $N_1^{th}$ MN, which is recruited at time $ft_{N_1}^1$. It non-physiologically yields $CSI(t < ft_{N_1}^1) = 0$. Accounting for this experimental limitation, $I(t)$ was defined to remain null until the first identified MN starts firing at $ft_{N_1}^1$, and to non-continuously reach $I_{N_1}^{th}$ at $t = ft_{N_1}^1$:

$$I(t) = \begin{cases} 0 \; if \; t < ft_{N_1}^1 \\ I_{N_1}^{th} + G \cdot CSI(t) else \end{cases} \text{with } G = \frac{I_{N_r}^{th} - I_{N_1}^{th}}{CSI(ft_{N_r}^1) - CSI(ft_{N_1}^1)} \qquad \text{(Eq 6)}$$

In (Eq 6), the rheobase currents of the first and last identified MNs $I_{N_1}^{th}$ and $I_{N_r}^{th}$ were estimated from a typical distribution of rheobase in a MN pool $I^{th}(j) = k_1 \cdot \Delta_I^{\left(\frac{j}{N}\right)^{k_3}}$, obtained as for the distribution of $F^{th}$, from normalized experimental data from populations of hindlimb alpha-MNs in adult rats and cats in vivo [84–92]. It is worth noting that the experimental $I^{th}$ were obtained by injecting current pulses directly into the soma of identified MNs, thus bypassing the dendritic activity and most of the aforementioned PIC-generated nonlinear mechanisms [40]. A $\Delta_I$ = 9.1-fold range in MN rheobase in [3.9; 35.5]$nA$ was taken [21,42], while a larger $\Delta_I$ is also consistent with the literature ([42], Table 4) and larger values of $I^{th}$ can be expected for humans [21].

## Step (3): LIF model – parameter tuning and distribution of electrophysiological properties in the MN pool

In the third step of the approach in Fig 1, single-compartment LIF MN models with current synapses are calibrated to mimic the discharge behaviour of the $N_r$ experimentally identified MNs.

## MN LIF model: description

A variant of the LIF MN model was chosen in this study for its relative mathematical simplicity and low computational cost and its adequacy in mimicking the firing behaviour of MNs. The LIF model described in (Eq 7) describes the discharge behaviour of a MN of rheobase $I^{th}$ and input resistance $R$ as a capacitor charging with a time constant $\tau$ and instantaneously discharging at time $ft$ when the membrane voltage $V_m$ meets the voltage threshold $V_{th}$, after which $V_m$ is reset and maintained to the membrane resting potential $V_{rest}$ for a time duration called

'inert period' *IP* in this study. For simplicity without loss of generalisation, the relative voltage threshold was defined as $\Delta V_{th} = V_{th} - V_{rest} > 0$, and $V_{rest}$ was set to 0.

The model is described by the following set of equations:

$$\begin{cases} \tau \dfrac{dV_m}{dt} = R \cdot I(t) - V_m \\ \tau = RC \\ V_m(ft) = \Delta V_{th} \\ \lim_{t \to ft^+} V_m = 0 \\ \forall t \in [ft; ft + IP], V_m = 0 \end{cases} \quad \text{(Eq 7)}$$

The differential equation was solved with a time step $dt \leq 0.001s$ as:

$$\begin{cases} V_m(0) = 0 \\ \forall n \in \mathbb{N}^*, V_m[nT] = e^{-\frac{T}{\tau}} V_m[(n-1)T] + R\dfrac{T}{\tau} \cdot I(nT) \end{cases} \quad \text{(Eq 8)}$$

This model includes 5 electrophysiological parameters: $R$, $C$, $\tau$, $\Delta V_{th}$ and *IP*. The passive parameters $R$ and $C$ were mathematically related to the MN surface area S as $R = \frac{k_R}{S^{2.43}}$ and $C = C_m \cdot S$ after an extensive meta-analysis of published experimental data on hindlimb alpha-MNs in adult cats in vivo [42], that sets the constant value of $C_m$ to $1.3 \mu F \cdot cm^2$, supports the equality $\tau = RC$ and the validity of Ohm's law in MNs as $I^{th} = \frac{2.7 \cdot 10^{-2}}{R} = \frac{\Delta V_{th}}{R}$, setting the constant value $\Delta V_{th} = 27mV$ in this study. The model was thus reduced to the MN size parameter $S$ and to the *IP* parameter.

## MN LIF model: the IP parameter

Single-compartment LIF models with no active conductances receiving current inputs can predict non-physiologically large values of MN firing frequency (FF) because of a linear FF-I gain at large but physiological current inputs $I(t)$ [44]. The saturation in the FF-I relation typically observed in the mammalian motor unit firing patterns is primarily mediated by the voltage-dependent activity of the PIC-generating dendritic channels [40], that was overlooked in the *CSI–I* transformation in Step (2), and which decreases the driving force of the synaptic current flow as the dendritic membrane depolarizes. While a physiological modelling of this saturation could be achieved with a LIF model with conductance synapses, as discussed in Limitation 3 in the Limitations section, this nonlinear behaviour could also be captured in a simple approach by tuning the phenomenological IP parameter in (Eq 7). Considering a constant supra-threshold current $I \gg I^{th}$ input to a LIF MN, the steady-state firing frequency FF predicted by the LIF model is:

$$FF(I) = \frac{1}{IP - RC\ln\left(1 - \frac{\Delta V_{th}}{RI}\right)} \underset{I \gg I^{th}}{\Longleftrightarrow} FF(I) \approx \frac{1}{IP + \frac{C \cdot \Delta V_{th}}{I}} \quad \text{(Eq 9)}$$

As $I$ and $C$ typically vary over a 10-fold and 2.4-fold range respectively [42], the FF predicted by the LIF model is dominantly determined by the value of *IP* as the input current increasingly overcomes the MN current threshold: $FF(I \gg I^{th}) \approx \frac{1}{IP}$.

In the previous phenomenological models of MNs [9,10,29], a maximum firing rate $FF^{max}$ was defined and a non-derivable transition from $FF(I)$ to constant $FF^{max}$ was set for increasing values of $I(t)$. Here, *IP* was integrated to the dynamics of the LIF model and was derived from

experimental data to be MN-specific in the following manner. For each of the $N_r$ identified MNs, the time-course of the MN instantaneous discharge frequencies (IDFs) was first smoothed, as performed in [55], with a sixth-order polynomial trendline ($IDF_{trend}$) to neglect any unexplained random noise in the analysis. The mean $M$ of the trendline values during the plateau of force $[t_{r_2}; t_{r_3}]$ was then obtained. If $\exists t \in [t_{r_1}; t_{r_2} - 1], IDF_{trend}(t) > 0.9M$, i.e. if the MN reached during the ramp of $I(t)$ in $[t_{r_1}; t_{r_2} - 1]$ an IDF larger than 90% of the IDF reached one second before the plateau of force, the MN was identified to 'saturate'. Its $IP$ parameter was set to $IP = \frac{1}{\max(IDF_{trend})}$, which constrains the MN maximum firing frequency for high input currents to $FF^{max} \approx \frac{1}{IP} = max(IDF_{trend})$. A power trendline $IP(j) = a \cdot j^b$ was finally fitted to the pairs $(N_i; IP_{N_i})$ of saturating MNs and the $IP_{N_i}$ values of the non-saturating MNs were predicted from this trendline. To account for the residual variation in FF observed to remain at high $I(t)$ due to random electrophysiological mechanisms, the $IP$ parameter was randomized at each firing time, taking the value $IP+o$, where $o$ was randomly obtained from a normal gaussian distribution of $\frac{IP}{10}$ standard deviation.

## MN LIF model: MN size parameter calibration

The remaining unknown parameter - the MN size $S$ - defines the recruitment and firing dynamics of the LIF model. The size $S_i$ of the $i^{th}$ identified MN was calibrated by minimizing over the time range $\left[t_{tr_0}; \frac{t_{tr_2}+t_{tr_3}}{2}\right]$ (Table 1) the cost function $J(S_i)$ computed as the root-mean-square difference between experimental $FIDF_i^{exp}(t)$ and LIF-predicted $FIDF_i^{sim}(t)$ filtered instantaneous discharge frequencies:

$$
\begin{cases}
J(S_i) = \sqrt{\dfrac{dt}{\frac{t_{tr_2}+t_{tr_3}}{2} - t_{r_0}} \cdot \sum_{t_k=t_{r_0}}^{\frac{t_{tr_2}+t_{tr_3}}{2}} \left(FIDF_i^{sim}(t_k) - FIDF_i^{exp}(t_k)\right)^2} \\
\min_{S_i} J(S_i), i \in [\![1; N_r]\!]
\end{cases}
\tag{Eq 10}
$$

To assess how well the calibrated LIF models can replicate the available experimental data, the normalized RMS error (nRMSE) (%) and coefficient of determination $r^2$ between $FIDF_i^{exp}(t)$ and $FIDF_i^{sim}(t)$, and the error in seconds between experimental and predicted recruitment times $\Delta ft_i^1 = ft_i^{1,sim} - ft_i^{1,exp}$ were computed for the $N_r$ MNs. Finally, a power trendline in (Eq 11) was fitted to the pairs $(N_i; S_{N_i})$, and the continuous distribution of MN sizes in the entire pool of $N$ MNs was obtained.

$$
S(j) = k_1 \cdot \Delta_s^{\left(\frac{j}{N}\right)^{k_3}}
\tag{Eq 11}
$$

$\Delta_S = 2.4$ was taken in [42]. The $S(j)$ distribution defines the continuous distribution of the MN-specific electrophysiological properties across the MN pool ([42], Table 4).

## MN LIF model: parameter identification during the derecruitment phase

The time-range $[t_{r_3}; t_{r_5}]$ over which the MNs are being derecruited was not considered in the $S$ calibration in (Eq 10) because the MN's current-voltage relation presents a hysteresis triggered by long-lasting PICs, as discussed in [40,93,94]. This hysteresis, which leads MNs to be derecruited at lower current threshold than at recruitment, can be phenomenologically interpreted in the scope of a single-compartment LIF model with no active conductance as an increase in the 'apparent' MN resistance $R$ during derecruitment to a new value $R^d$. To

determine $R^d$, a linear trendline $I^{dth} = k_{th}^{dth} \cdot I^{th}$ ($k_{th}^{dth} < 1$) was fitted to the association of experimental MN recruitment $I^{th}$ and recruitment $I^{dth}$ current thresholds for the $N_r$ recruited MNs, and the distribution of MN input resistance $R = \frac{k_R}{S^{2.43}}$ was increased over the derecruitment time range $[t_{r_3}; t_{r_5}]$ to:

$$R^d = \frac{k_R^d}{S^{2.43}} \text{ with } k_R^d = \frac{k_R}{k_{th}^{dth}} > k_R \qquad (\text{Eq 12})$$

As reviewed in [20,40], the current-voltage hysteresis also explains the typically lower MN discharge rate observed at derecruitment than at recruitment. For purely modelling perspectives, this phenomenon was phenomenologically captured by increasing over the time-range $[t_{r_3}; t_{r_5}]$ the value of the $C_m$ property, which is the main parameter influencing the precited MN FF in (Eq 7) for current inputs close to $I^{th}$. The simulated $FIDF_i^{sim}$ traces were iteratively simulated for the $N_r$ MNs over the time-range $[t_{r_3}; t_{r_5}]$ with the $N_r$ $S_i$–$IP_i$–calibrated LIF models obtained from Step (3), for $0.1 \mu F \cdot cm^2$ incremental changes in the value of the membrane specific capacitance $C_m$. The $FIDF_i^{sim}$ results were compared to the $FIDF_i^{exp}$ traces with nRMSE and $r^2$ values. The 'apparent' $C_m$ value returning the lowest output for (Eq 13) was retained and was renamed $C_m^d$.

$$J(C_m) = \frac{\frac{\overline{nRMSE}}{100} - \overline{r^2}}{2} \qquad (\text{Eq 13})$$

In the following, the individual spike trains $sp_i^{sim}(t)$ were predicted with $C_m$ over the $[t_{r_0}; t_{r_3}[$ time range and $C_m^d$ over the $[t_{r_3}; t_{r_5}]$ time range.

It must be noted that, although this approach can accurately capture the observed MN nonlinear input-output behaviour and is coherent with the modelling constraints imposed by the LIF formulation in (Eq 7), as the $C_m$ and $R$ parameters mainly affect the discharge and recruitment properties of the LIF model respectively in this model, the $R$ and $C_m$ parameters are passive MN properties which remain independent from the MN synaptic and membrane activity in actual MNs, while the $R$-to-$R^d$ and $C_m$-to-$C_m^d$ transformations are merely phenomenological interpretations of complex neurophysiological mechanisms, and do not provide any insights into the working principles underlying MN recruitment. This limitation is further discussed in Limitation 3.

## Step (4): Simulating the MN pool firing behaviour

The firing behaviour of the complete MN pool, i.e. the MN-specific spike trains $sp_j(t)$ of the $N$ virtual MNs constituting the MN pool, was predicted with a cohort of $N$ LIF models receiving the common synaptic current $I(t)$ as input. The inert period $IP_j$ and MN size $S_j$ parameters scaling each LIF model were obtained from the distributions $IP(j)$ and $S(j)$ previously derived at Step (3).

## Validation

**Validation 1.** We assessed whether the $N_r$ experimental spike trains $sp_i^{exp}(t)$ were accurately predicted by this 4-step approach (Fig 1). Steps (2) and (3) (Fig 1) were iteratively repeated with samples of $N_r$–1 spike trains, where one of the experimentally recorded MNs was not considered. At each i[th] iteration of the validation, the i[th] identified spike train $sp_i^{exp}(t)$ was not used in the derivation of the synaptic current $I(t)$ (step 2) and in the reconstruction of the MN size and $IP$ distributions $S(j)$ and $IP(j)$ in the MN pool (step 3). As in Step (4), the

$FIDF_i^{sim}$ of the $i^{th}$ MN was finally predicted with a LIF model, which was scaled with the parameters $S(N_i)$ and $IP(N_i)$ predicted from the $S(j)$ and $IP(j)$ distributions. For validation, $FIDF_i^{sim}$ was compared to the experimental $FIDF_i^{exp}$ with calculation of $\Delta ft_i^1$, $r^2$ and nRMSE values. This validation was iteratively performed for all the $N_r$ identified MN spike trains.

**Validation 2.** We assessed whether the $N$ MN spike trains $sp_j^{sim}(t)$, predicted in Step 4 for the entire MN pool from the $N_r$ identified trains $sp_i^{exp}(t)$, accurately predicted the effective neural drive (eND$_N$) to the muscle. The eND$_N$ was computed as the $[0;4]$Hz low-pass filtered cumulative spike train of the $N$ predicted spike trains, $CST_N = \sum_{j=1}^{N} sp_j^{sim}(t)$. As suggested for isometric contractions [83], the normalized $eND_N$ was compared for validation against the normalized experimental force trace $\overline{F}(t)$ with calculation of the nRMSE and $r^2$ metrics. $\overline{F}(t)$ was also compared with nRMSE and $r^2$ to the normalized effective neural drive $eND_{N_r}$ computed directly from the $N_r$ experimentally identified MN spike trains. The added value of the presented workflow (Steps 1 to 4) in predicting the neural drive to muscle was finally assessed by comparing the (nRMSE, $r^2$) pairs obtained for the $eND_{N_r}$ and $eND_N$ traces. Considering the uncertainty on the value of $N$ in the literature, as previously discussed, the performance of the method with reconstructed populations of $N = \{N_r, 100, 200, 300, 400\}$ MNs was investigated. This sensitivity analysis provides different scaling factors $N$ for the normalized MN mapping (Step 1) and $IP$ and $S_{MN}$ parameter distributions (Step 3) and constrains the number of firing MNs involved in the computation of eND$_N$ (Step 4).

### Application to MN-driven muscle modelling

The $N$ MN-specific spike trains $sp_j^{sim}(t)$ predicted in step (4) were input to a phenomenological muscle model to predict the whole muscle force trace $F_N^{sim}$ in a forward simulation of muscle voluntary isometric contraction. As displayed in Fig 3, the muscle model was built as $N$ in-parallel Hill-type models which were driven by the simulated spike trains $sp_j^{sim}(t)$ and replicated the excitation-contraction coupling dynamics and the contraction dynamics of the $N$ MUs constituting the whole muscle. In brief, the binary spike train $sp_j^{sim}(t)$ triggered for each $j^{th}$ MU the trains of nerve and muscle fibre action potentials that drove the transients of calcium ion $Ca^{2+}$ and $Ca^{2+}$-troponin complex concentrations in the MU sarcoplasm, yielding in a last step the time-history of the MU active state $a_j(t)$. The MU contraction dynamics were reduced to a normalized force-length relationship that scaled nonlinearly with the MU active state [95] and transformed $a_j(t)$ into a normalized MU force trace $\overline{f}_j(t)$. The experiments being performed at

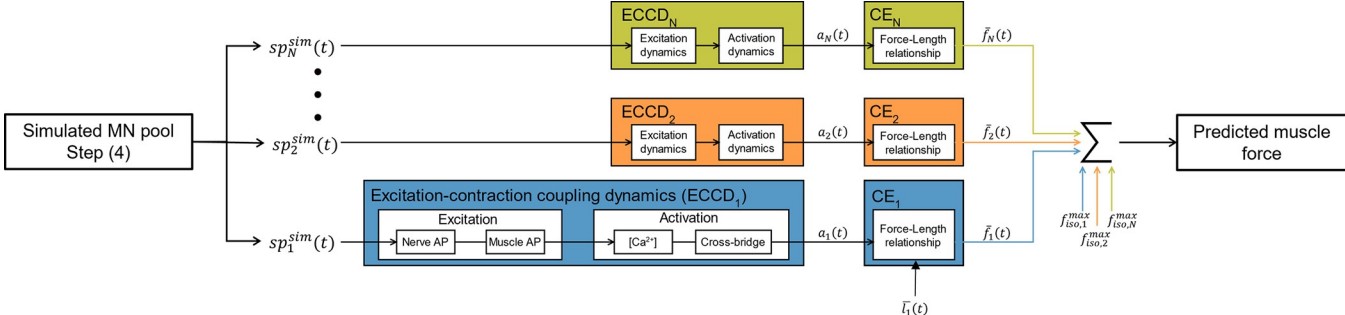

**Fig 3. MN-driven neuromuscular model.** The N in-parallel Hill-type models take as inputs the N spike trains sp$_j^{sim}$(t) predicted in steps (1-4) and output the predicted whole muscle force trace F$_m^{sim}$(t). The MU-specific active states a$_j$(t) are obtained from the excitation-contraction coupling dynamics adapted from [98] and extended to model the concentration of the Calcium-troponin system. The MU normalized forces $\overline{f}_j$(t) are computed by the MU contractile elements (CE$_i$) at MU optimal length and are scaled with values of MU maximum isometric forces f$_{iso,j}^{max}$ to yield the MU force traces f$_j$(t). The predicted whole muscle force is taken as F$_N^{sim}$(t) = $\sum_j$f$_j$(t).

constant ankle joint (100°, 90° being ankle perpendicular to tibia) and muscle-tendon length, it was simplified, lacking additional experimental insights, that tendon and fascicle length both remained constant during the whole contractile event at optimal MU length $\overline{l_{opt,j}} = 1$. The dynamics of the passive muscle tissues and of the tendon and the fascicle force-velocity relationships were therefore neglected. Finally, the MU-specific forces $f_j(t)$ were derived with a typical muscle-specific distribution across the MU pool of the MU isometric tetanic forces $f_{iso}^{max}(j)$ [64,71,74]. The whole muscle force was obtained as the linear summation of the MU forces $F_N^{sim}(t) = \sum_j f_j(t)$.

To validate $F_N^{sim}(t)$, the experimental muscle force $F^{exp}(t)$ was first approximated from the experimental force trace $F(t)$, which was recorded at the foot with a force transducer [26]. The transducer-ankle joint and ankle joint-tibialis anterior moment arms $L_1$ and $L_2$ were estimated using OpenSim [96] and a generic lower limb model [97]. Using the model muscle maximum isometric forces, it was then inferred the ratio $q$ of transducer force $F(t)$ that was taken at MVC by the non-TA muscles spanning the ankle joint in MVC conditions. The experimental muscle force was estimated as $F^{exp}(t) = (1-q) \cdot \frac{L_1}{L_2} \cdot F(t)$ and was compared with calculation of normalized maximum error (nME), nRMSE and $r^2$ values against the muscle force $F_N^{sim}(t)$ predicted by the MN-driven muscle model from the $N$ neural inputs.

The whole muscle force $F_{N_r}^{sim}(t)$ was also predicted using the $N_r$ experimental spike trains $sp_i^{exp}(t)$ as inputs to the same muscle model of $N_r$ in-parallel Hill-type models (Fig 3). In this case, each normalized MU force trace $\overline{f_j}(t)$ was scaled with the same $f_{iso}^{max}(j)$ distribution, however assuming the $N_r$ MNs to be evenly spread in the MN pool. $F_{N_r}^{sim}(t)$ was similarly compared to $F^{exp}(t)$ with calculation of $nME$, $nRMSE$ and $r^2$ values. To assess the added value of the step (1-4) approach in the modelling of MN-driven muscle models, the (nME, nRMSE, $r^2$) values obtained for the predicted $F_{N_r}^{sim}(t)$ and $F_N^{sim}(t)$ were compared.

## Results

### Experimental data

As reported in Table 1, the experimental datasets $DTA_{35}$ and $HTA_{35}$ respectively identified 32 and 21 spike trains $sp_i^{exp}(t)$ from the trapezoidal isometric TA muscle contraction up to 35% MVC, $HTA_{50}$ identified 14 $sp_i^{exp}(t)$ up to 50%MVC, and $HGM_{30}$ identified 27 $sp_i^{exp}(t)$ from the GM muscle up to 30%MVC. The $N_r = 32$ MN spike trains, identified in this dataset across the complete TA pool of $N = 400$ MNs, are represented in Fig 4A in the order of increasing force recruitment thresholds $F_i^{th}$. The $N_r$ MNs were globally derecruited at relatively lower force thresholds (Fig 4B) and generally discharged at a relatively lower firing rate (Fig 4C) at derecruitment than at recruitment.

### Step (1): MN mapping

The $N_r = 32$ MNs identified in the dataset $DTA_{35}$ were allocated to the entire pool of $N = 400$ MNs according to their recruitment thresholds $F_i^{th}$ (%MVC). The typical TA-specific frequency distribution of the MN force recruitment thresholds $F^{th}$, which was obtained from the literature and reported in the bar plot in Fig 5A, was approximated (Fig 5B) by the continuous relationship:

$$F^{th}(j) = 0.50 \cdot \left( 58.12 \cdot \frac{j}{N} + \Delta_F^{\left(\frac{j}{N}\right)^{1.83}} \right), j \in [1; N] \tag{Eq 14}$$

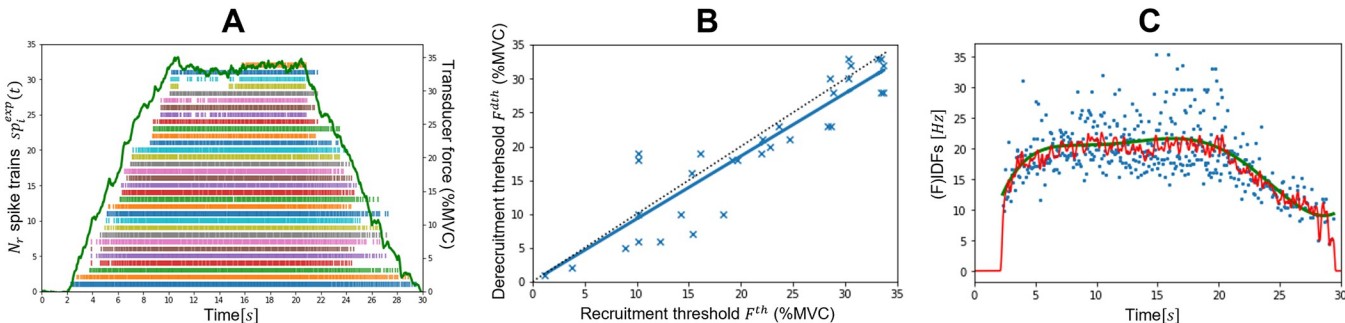

**Fig 4. Experimental data obtained from HDEMG signal decomposition in the dataset $DTA_{35}$.** (A) Time-histories of the transducer force trace in %MVC (green curve) and of the $N_r$ = 32 MN spike trains identified from HDEMG decomposition and ranked from bottom to top in the order of increasing force recruitment thresholds $F^{th}$. (B) Association between force recruitment and derecruitment threshold, fitted by a linear trendline $y = 0.9 \cdot x$ ($r^2 = 0.85$). Identity is displayed as a black dotted line. (C) Time-histories of the MN instantaneous discharge frequencies (IDFs, blue dots) smoothed by convolution with a Hanning window of length 400ms (red curve) and with a sixth-order polynomial trendline (green curve) of the lowest-threshold identified ($1^{st}$) MN.

With this distribution, 231 TA MNs, i.e. 58% of the MN pool is recruited below 20%MVC, which is consistent with previous conclusions [20].

From the $F^{th}(j)$ distribution, the $N_r$ identified MNs were mapped to the complete MN pool (blue crosses in Fig 5B) according to their recorded force recruitment thresholds $F^{th}_i$ (ordinates in Fig 4B). As shown in Fig 5C, the $N_r$ MNs identified experimentally were not homogeneously spread in the entire MN pool ranked in the order of increasing force recruitment thresholds, as two MNs fell in the first quarter of the MN pool, 5 in the second quarter, 18 in the third quarter and 5 in the fourth quarter. Such observation was similarly made in the three other experimental datasets, where no MN was identified in the first quarter and in the first half of the MN pool in the datasets $HGM_{30}$ and $HTA_{50}$ respectively (second column of Table 2). In all four datasets, mostly high-thresholds MNs were identified experimentally.

## Step (2): Common current input $I(t)$

To approximate the common synaptic input $CSI(t)$ to the MN pool, the CST and the eND to the MN pool were obtained in Fig 6 from the $N_r$ MN spike trains identified experimentally using (Eq 4) and (Eq 5). After 20 random permutations of $\frac{N_r}{2}$ complementary populations of MNs, an average coherence of $coher_{\frac{N_r}{2},mean} = 0.56$ was obtained between $\frac{N_r}{2}$-sized CSTs of the $DTA_{35}$ dataset. From Fig 2 in [47], a coherence of $coher_{N_r} = 0.8$ is therefore expected between the CST in Fig 6A and a typical CST obtained with another virtual group of $N_r$ = 32 MNs, and by extension with the true CST obtained with the complete MN pool. The normalized eND and force traces (black and green curves respectively in Fig 6B) compared with $r^2 = 0.92$ and $nRMSE = 20.0\%$ for the $DTA_{35}$ dataset. With this approach, we obtained $coher_{N_r} > 0.7$, $r^2 > 0.7$ and $nRMSE < 30\%$ for all four datasets, with the exception of $HGM_{30}$ for which $coher_{N_r} < 0.7$ (third column of Table 2). For the TA datasets, the sample of $N_r$ identified MNs was therefore concluded to be sufficiently representative of the complete MN pool for its linearity property to apply, and the eND (red curve in Fig 6B for the dataset $DTA_{35}$) in the bandwidth $[0,10]Hz$ was confidently identified to be the common synaptic input ($CSI$) to the MN pool. As observed in Fig 6B, a non-negligible discrepancy, which partially explains why $coher_{N_r} \neq 1$, was however systematically obtained between the eND and force traces in the regions low forces, where mostly small low-threshold MNs are recruited. As discussed in the Discussion Section, this discrepancy reflects the undersampling of small MUs identified from decomposed HDEMG signals and a bias towards mainly identifying the large high-threshold units which are recruited

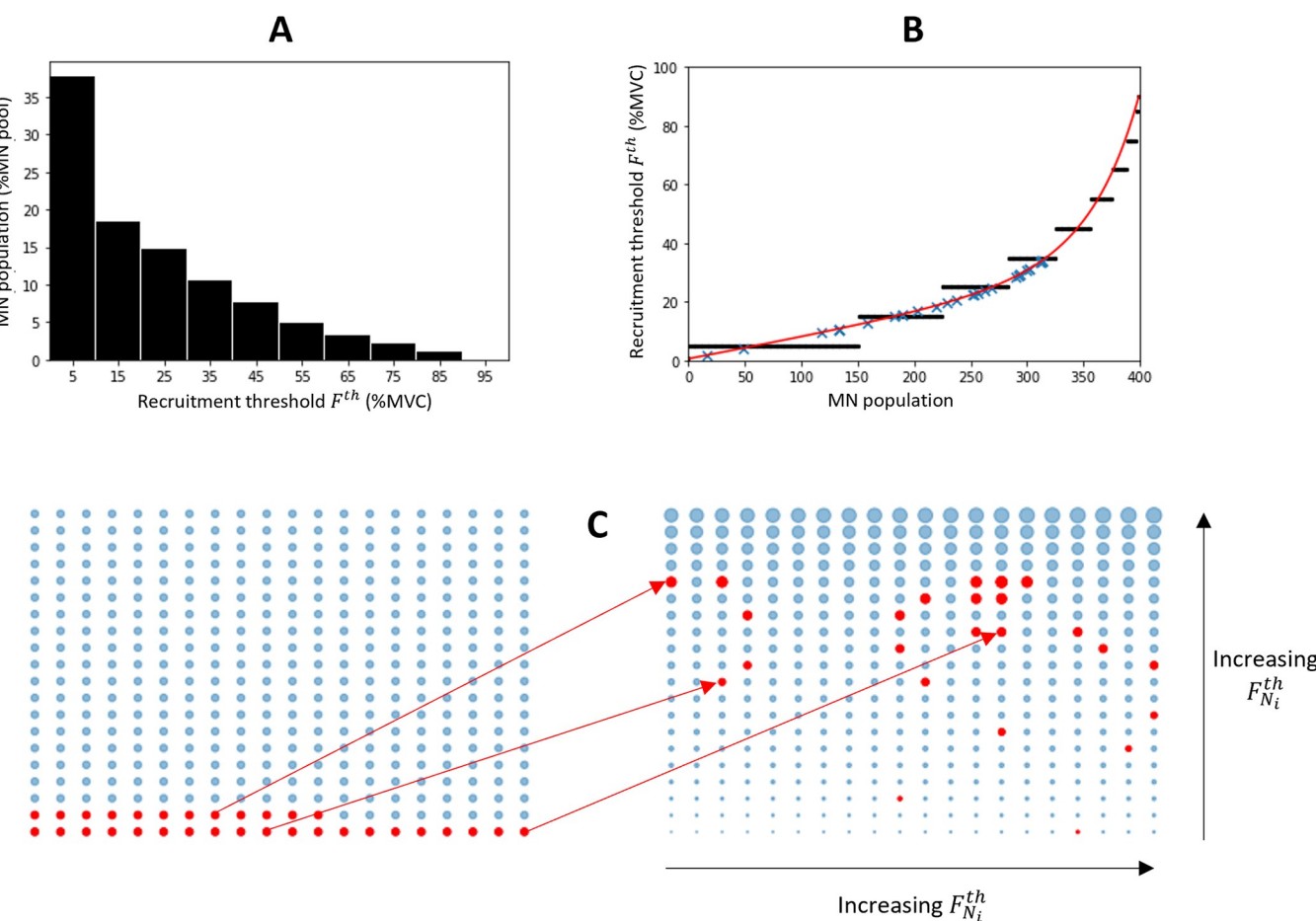

**Fig 5.** Distribution of force recruitment thresholds $F^{th}$ in the human Tibialis Anterior (TA) muscle and mapping of the $N_r$ identified MNs to the complete MN pool. (A) Typical partition obtained from the literature of the TA MN pool in 10% increments in normalized $F^{th}$. (B) Equivalent $F^{th}$ stepwise distribution (black dots) in a TA pool of N = 400 MNs, approximated by the continuous relationship $F^{th}(j)$ (red curve). The mapping (blue crosses) of the $N_r$ = 32 MNs identified in the dataset DTA$_{35}$ was obtained from the recorded $F_i^{th}$ (Fig 4B). (C) $N_r$ = 32 MNs (red dots) of unknown properties (Left) are mapped (Right) to the complete MN pool from the $F^{th}(j)$ distribution, represented by the blue dots of increasing sizes. The MNs represented here are numbered from left to right and bottom to top from j = 1 to 400.

**Table 2. Intermediary results obtained for the datasets DTA$_{35}$, HTA$_{35}$, HTA$_{50}$ and HGM$_{30}$ from the three first steps of the approach.** For each dataset are reported (1) the locations in the complete pool of N MNs of the lowest-($N_1$) and highest-threshold ($N_{N_r}$) MNs identified experimentally, (2) the coher$_{N_r}$ value between the experimental and virtual cumulative spike trains (CST), and the coefficients defining the distributions in the complete MN pool of (3) the inert period (IP) parameter and of (4) the MN size (S). For the TA and GM muscles, N = 400 and N = 550 respectively.

| | Identified population | | CST | $IP(j)[s] = a \cdot j^b$ | | $S(j)[m^2] = S_{min} \cdot \Delta_S^{\left(\frac{j}{N}\right)^c}$ | |
|---|---|---|---|---|---|---|---|
| Dataset | $N_r$ | $i \rightarrow N_i$ | $coher_{N_r}$ | $a$ | $b$ | $S_{min}[m^2]$ | $c$ |
| $DTA_{35}$ | 32 | 9-313 | 0.80 | 0.04 | 0.06 | $1.49 \cdot 10^{-7}$ | 1.47 |
| $HTA_{35}$ | 21 | 72-313 | 0.71 | 0.04 | 0.05 | $1.73 \cdot 10^{-7}$ | 2.76 |
| $HTA_{50}$ | 14 | 233-346 | 0.71 | 0.0006 | 0.80 | $1.35 \cdot 10^{-7}$ | 0.85 |
| $HGM_{30}$ | 27 | 162-412 | 0.65 | 0.03 | 0.20 | $1.20 \cdot 10^{-7}$ | 0.57 |

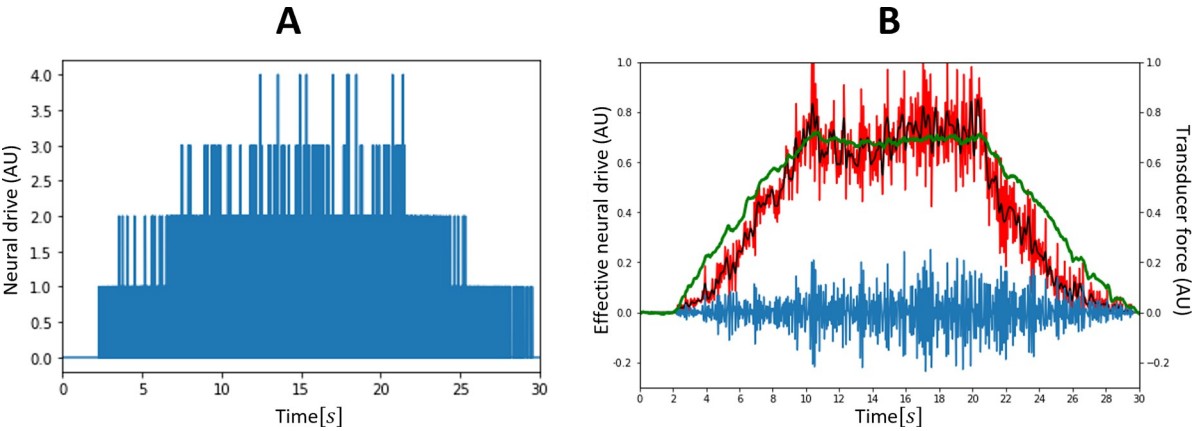

**Fig 6. Neural drive to the muscle derived from the $N_r$ identified MN spike trains in the dataset $DTA_{35}$.** (A) Cumulative spike train (CST) computed by temporal binary summation of the $N_r$ identified MN spike trains. (B) Effective neural drive – Upon applicability of the linearity properties of subsets of the MN pool, the effective neural drive is assimilated as the synaptic input to the MN pool. The normalized common synaptic input (red), control (black) and noise (blue) are obtained from low-pass filtering the CST in the bandwidths relevant for muscle force generation. The normalized experimental force trace (green curve) is displayed for visual purposes.

close to the plateau of force. From Fig 2 in [47], it is worth noting that the computed *CSI* in Fig 6B (red curve) accounts for 60% of the variance of the true synaptic input, which is linearly transmitted by the MN pool, while the remaining variance is the MN-specific synaptic noise, which is assumed to be filtered by the MN pool and is neglected in the computation of the *eND* in this workflow.

As discussed in the Methods section, the normalized *CSI* (red curve in Fig 6B) was simplified to be linearly related to the total dendritic membrane current $I(t)$. To do so, the scaling factor was determined with a typical distribution of the MN membrane rheobase $I^{th}(j)$ in a cat MN pool, which was obtained (Fig 7A) from the literature [85,87,88,90,91,99–101] as:

$$I^{th}(j) = 3.9 \cdot 10^{-9} \cdot \Delta_I^{\left(\frac{j}{N}\right)^{1.18}}, j \in [1;N] \tag{Eq 15}$$

Using (Eq 6), the time-history of the current input $I(t)$ was obtained (Fig 7B):

$$I(t) = \begin{cases} 0 \ if \ t < ft^1_{N_1} \\ 3.9 \cdot 10^{-9} + 6.0 \cdot 10^{-8} \cdot CSI(t) \ else \end{cases} \tag{Eq 16}$$

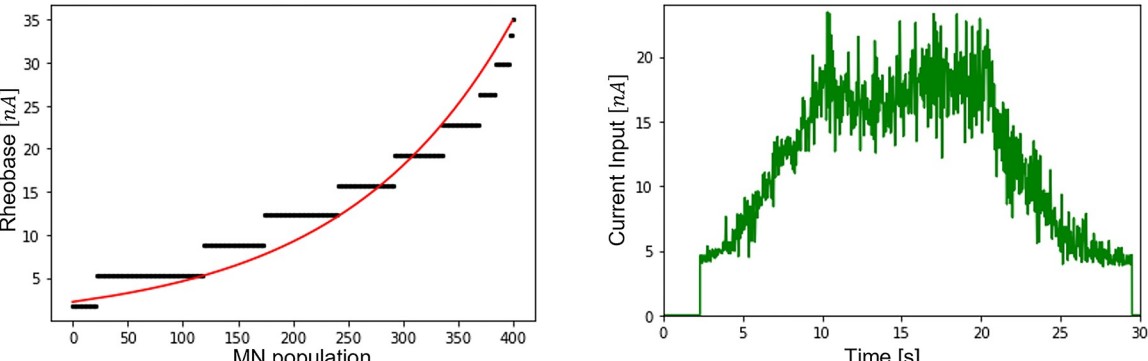

**Fig 7.** (A) Typical distribution of MN current recruitment threshold $I^{th}$ ($N_i$) in a cat MN pool according to the literature. (B) Current input I(t), taken as a non-continuous linear transformation of the common synaptic input (red curve in Fig 6B) for the dataset $DTA_{35}$.

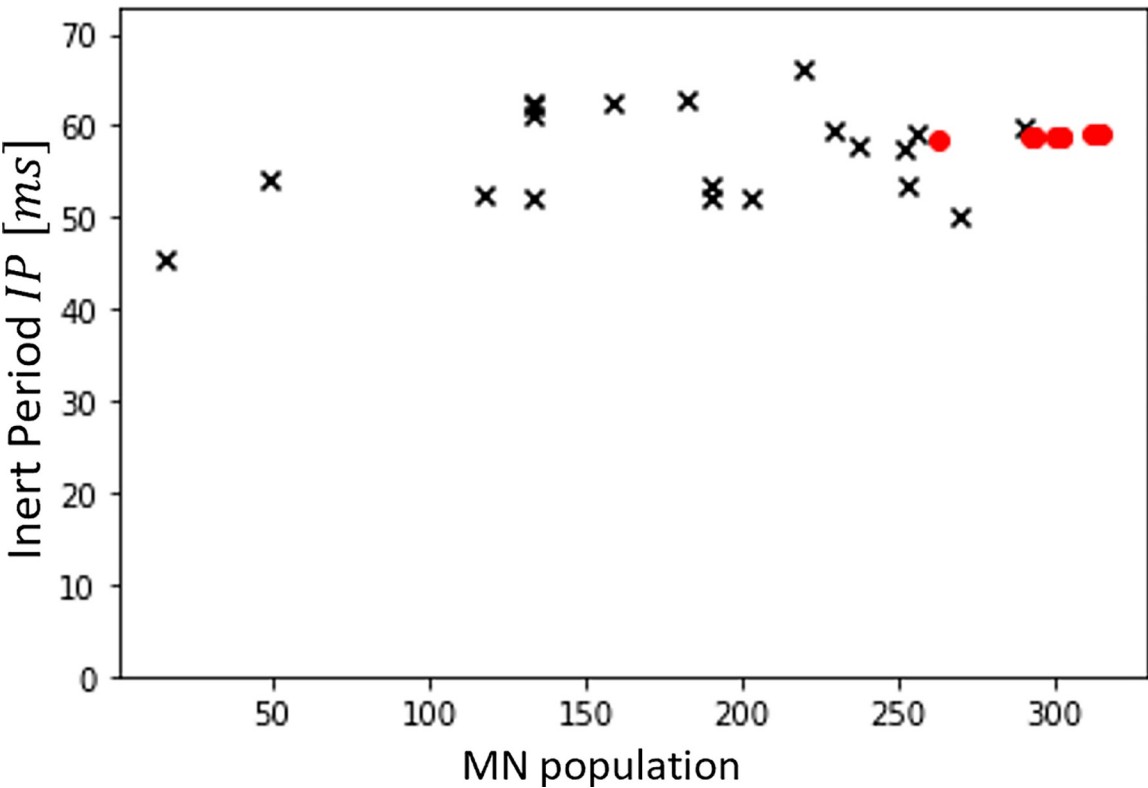

**Fig 8.** MN Inert Periods (IPs) in ms obtained from the experimental measurements of IDFs in dataset $DTA_{35}$. The twenty lowest-threshold MNs are observed to 'saturate' as described in the Methods and their IP (black crosses) is calculated as the inverse of the maximum of the trendline fitting the time-histories of their instantaneous firing frequency. The IPs of the 12 highest-threshold MNs (red dots) are obtained by trendline extrapolation.

## Step (3): LIF model – MN size calibration and distribution

Because of the modelling choices made for our MN LIF model, the MN inert period (*IP*) and the MN size *S* parameters entirely define the LIF-predictions of the MN firing behaviour. The $IP_i$ parameters of the $N_r$ MNs in the dataset $DTA_{35}$ (Fig 8) were obtained from the maximum firing frequency of the 20 MNs identified to 'saturate', from which the distribution of *IP* values in the entire pool of *N* MNs was obtained: $IP(j)[s] = 0.04 \cdot j^{0.05}$. With this approach, the maximum firing rate assigned to the first recruited and unidentified MN is $\frac{1}{IP_1} = \frac{1}{0.04s} = 25 Hz$. The *IP* distributions obtained with this approach for the three other datasets are reported in the fourth column of Table 2 and yielded physiological approximations of the maximum firing rate for the unidentified lowest -threshold MN for all datasets, with the exception of the dataset $HTA_{50}$, which lacks the information of too large a fraction of the MN pool for accurate extrapolations to be performed.

The size parameter $S_i$ of the $N_r$ LIF models was calibrated using the minimization function in (Eq 14) so that the LIF-predicted filtered discharge frequencies $FDIF_i^{sim}(t)$ of the $N_r$ MNs replicated the experimental $FDIF_i^{exp}(t)$, displayed in Fig 9A and 9B (blue curves). As shown in Fig 9C, the recruitment time $ft^1$ of three quarters of the $N_r$ identified MNs was predicted with an error less than 250ms. The calibrated LIF models were also able to accurately mimic the firing behaviour of the 27 lowest-threshold MNs as experimental and LIF-predicted FIDF traces compared with $r^2 > 0.8$ and $nRMSE < 15\%$ (Fig 9D and 9E). The scaled LIF models reproduced the firing behaviour of the five highest-threshold MNs ($Ni > 300$) with moderate accuracy, with

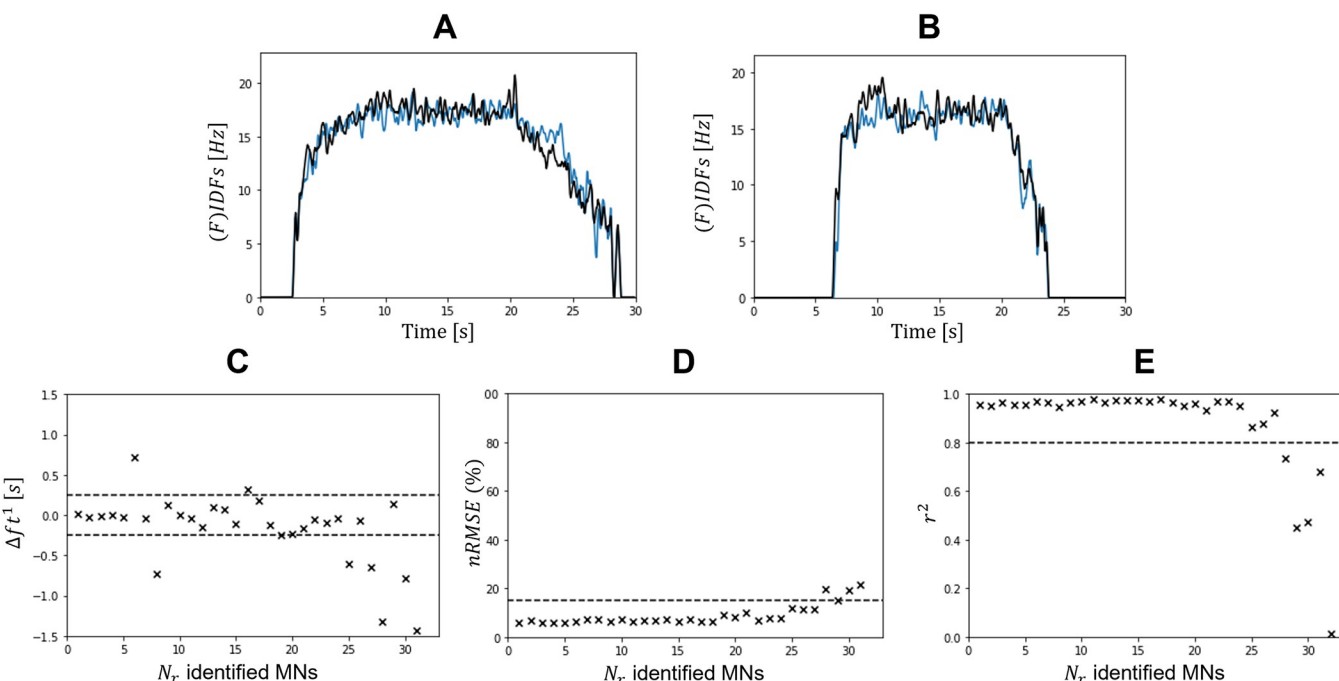

**Fig 9. Calibration of the MN size $S_i$ parameter.** (A and B) Time-histories of the experimental (black) versus LIF-predicted (blue) filtered instantaneous discharge frequencies (FIDFs) of the 2nt (A) and 16th (B) MNs identified in the $DTA_{35}$ dataset after parameter calibration. (C) Absolute error $\Delta ft^1$ in seconds in predicting the MN recruitment time with the calibrated LIF models. The accuracy of LIF-predicted FIDFs is assessed for each MN with calculation of the nRMSE (D) and $r^2$ (E) values. (C-E) The metrics are computed for the $[t_{r_0}; t_{r_3}]$ time range only. The dashed lines represent the $\Delta ft^1 \in [-250; 250]ms$, $nRMSE \in [0; 15]\%MVC$ and $r^2 \in [0.8, 1.0]$ intervals of interest respectively.

$\Delta ft^1$ and *nRMSE* values up to -1.5s and 18.2%. Despite modelling limitations discussed in the Discussion section, the global results in Fig 9 confirm that the two-parameters calibrated LIF models can accurately reproduce the firing and recruitment behaviour of the $N_r$ experimental MNs. It must be noted that the $\Delta ft^1$ and $r^2$ metrics were not included in the calibration procedure and the $(\Delta ft^1, r^2)$ values reported in Fig 9 were therefore blindly predicted.

The cloud of $N_r$ pairs $\{N_i; S_{N_i}\}$ of data points (black crosses in Fig 10A), obtained from MN mapping (Fig 5B and 5C) and size calibration for the dataset $DTA_{35}$, were least-squares fitted (red curve in Fig 10A, $r^2 = 0.96$) by the power relationship in (Eq 17), with $\Delta_S = 2.4$:

$$S(j) = 1.49 \cdot 10^{-7} \cdot \Delta_S^{\left(\frac{j}{N}\right)^{1.47}} \tag{Eq 17}$$

As reported in the last column of Table 2, the minimum MN size (in $m^2$) obtained by extrapolation of this trendline was in the range $[1.20, 1.73] \cdot 10^{-1} mm^2$ for the datasets $DTA_{35}$, $HTA_{35}$ and $HGM_{30}$, with expected maximum MN size (in $m^2$) in the range $[2.88, 4.15] \cdot 10^{-1} mm^2$ which is consistent with typical cat data [42,102–107], even if slightly in the lower range. The low value for the $c$ coefficient observed in Table 2 for the $TA_{50}$ and $GM_{30}$ datasets suggests that the typical skewness in MN size distribution reported in the literature [42] was not captured for these two datasets, in which low-threshold MNs in the first quarter of the MN population were not identified from HDEMG signals.

As displayed in Fig 9A and 9B, the phenomenological adjustment of the $R$ and $C_m$ parameters described in (Eq 12) and (Eq 13) successfully captured the main effects of the hysteresis mechanisms involved with MN derecruitment, where MNs discharge at lower rate and stop

  

firing at lower current input than at recruitment. In all four datasets, $J(C_m) = \overline{\frac{\overline{nRMSE} - \overline{r^2}}{2}}$ was parabolic for incremental values of $C_m$ and a global minimum was found for $C_m^d = 2.0, 2.0, 1.6$ and $2.2\mu F \cdot cm^2$ for the datasets $DTA_{35}$, $HTA_{35}$, $HTA_{50}$ and $HGM_{30}$, respectively. These values for the parameter $C_m^d$ were retained for the final simulations (Step (4)) over the $[t_{r_3}; t_{r_5}]$ time range.

### Step (4): Simulating the MN pool firing behaviour

As displayed in Fig 10B for the dataset $DTA_{35}$, the $S(j)$ distribution determined the MN-specific electrophysiological parameters (input resistance $R$ and membrane capacitance $C$) of a cohort of $N = 400$ LIF models, which predicted from $I(t)$ the spike trains of the entire pool of $N$ MNs (Fig 10C). As displayed in the plot of smoothed FIDFs in Fig 10C, the output behaviour of the reconstructed MN pool agrees with the Onion-skin scheme [108] where lower-threshold MNs start discharging at higher frequencies and reach higher discharge rates than larger recruited units.

### Validation

The four-step approach summarized in Fig 1 is detailed in Fig 11 and was validated in two ways.

**Validation 1.** The simulated spike trains were validated for the $N_r$ MNs by comparing experimental and LIF-predicted FIDFs, where the experimental information of the investigated MN was removed from the experimental dataset and not used in the derivation of the $IP$ $(j)$ and $S(j)$ distributions of the synaptic current $I(t)$. For the $N_r$ MNs of each dataset, Fig 12 reports the absolute error $\Delta ft^1$ in predicting the MN recruitment time (1st row) and the comparison between experimental and LIF-predicted filtered instantaneous discharge frequencies (FIDFs) with calculation of $nRMSE$ (%) (2nd row) and $r^2$ (3rd row) values over the $[t_{r_0}; t_{r_5}]$ time range. In all datasets, the recruitment time of 45-to-65% of the $N_r$ identified MNs was predicted with an absolute error less than $\Delta ft^1 = 250ms$. In all datasets, the LIF-predicted and experimental FIDFs of more than 80% of the $N_r$ MNs compared with $nRMSE < 20\%$ and $r^2 > 0.8$, while 75% of the $N_r$ MN experimental and predicted FIDFs compared with $r^2 > 0.8$ in the dataset $HGM_{30}$. These results confirm that the four-step approach summarized in Fig 1 is valid for all four datasets for blindly predicting the recruitment time and the firing behaviour of the $N_r$ MNs recorded experimentally. The identified MNs that are representative of a large fraction of the complete MN pool, i.e. which are the only identified MN in the range $\left[ N_i - \frac{N}{10}; N_i + \frac{N}{10} \right]$ of the entire MN pool, such as the 1st MN in the dataset $DTA_{35}$ (Fig 5C), are some of the MNs returning the highest $\Delta ft^1$ and $nRMSE$ and lowest $r^2$ values. As observed in Fig 12, ignoring the spike trains $sp_i^{exp}(t)$ of those 'representative' MNs in the derivation of $I(t)$, $IP(j)$ and $S(j)$ in steps (2) and (3) therefore affects the quality of the predictions more than ignoring the information of MNs that are representative of a small fraction of the MN population. In all datasets, $nRMSE > 20\%$ and $r^2 < 0.8$ was mainly obtained for the last-recruited MNs (4th quarter of each plot in Fig 12) that exhibit recruitment thresholds close to the value of the current input $I(t)$ during the plateau of constant force in the time range $[t_{tr_2}; t_{tr_3}]$. In all datasets, the predictions obtained for all other MNs that have intermediate recruitment thresholds and are the most identified MNs in the datasets, were similar and the best among the pool of $N_r$ MNs.

**Validation 2.** The effective neural drive predicted by the 4-step approach summarized in Fig 1 was validated by comparing for isometric contractions, the normalized $eND_N$ (orange traces in Fig 13) computed from the $N$ predicted spike trains $sp_j^{sim}(t)$ to the normalized force

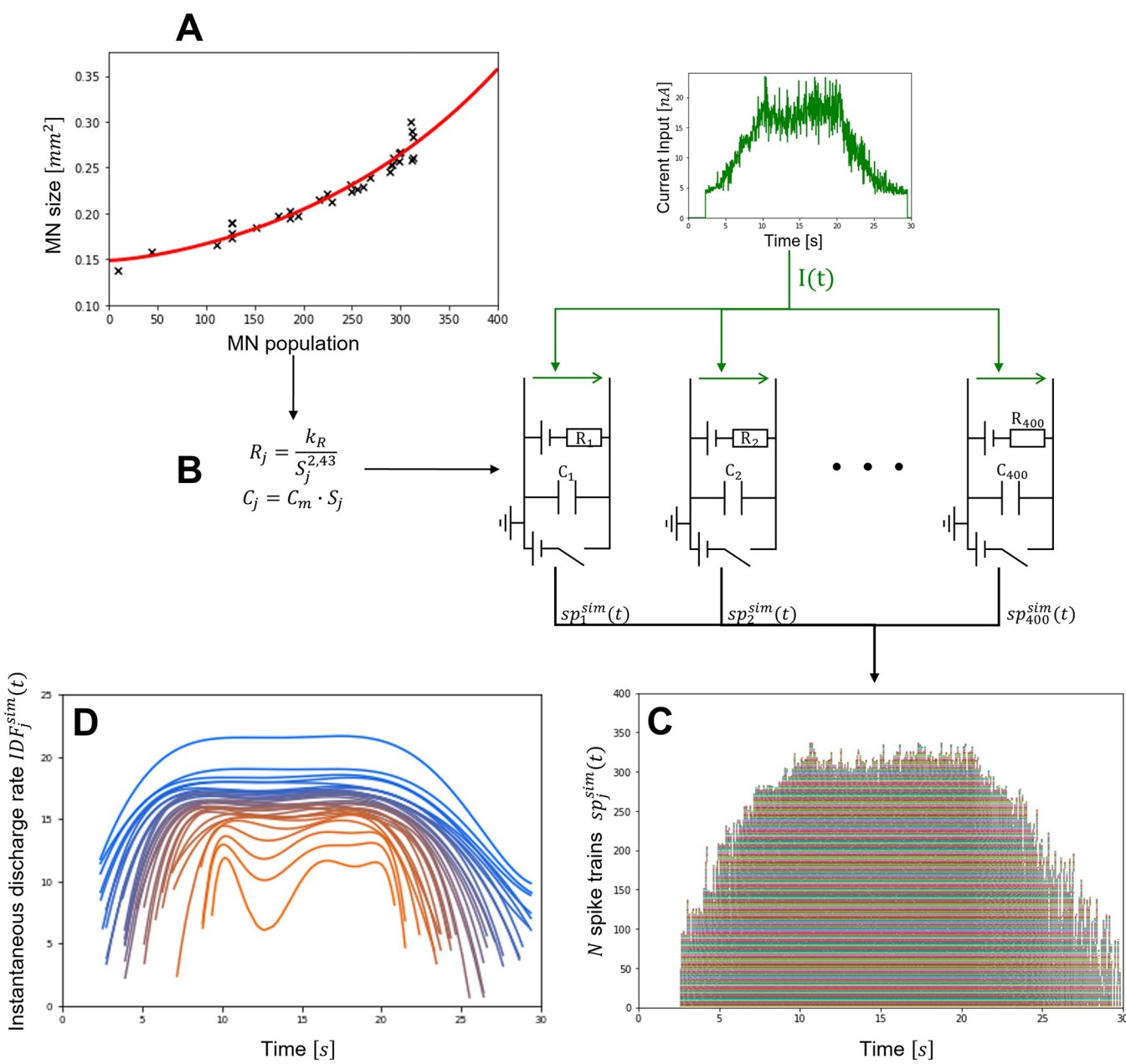

**Fig 10. Reconstruction of the firing behaviour and recruitment dynamics of the complete MN pool.** (A) The $N_r$ calibrated MN sizes (black crosses) are lest-squares fitted by the power trendline $S(j)[m^2] = 1.49 \cdot 10^{-7} \cdot \Delta_S^{\left(\frac{j}{N}\right)^{1.47}}$, which reconstructs the distributions of MN sizes in the complete MN pool. (B) The $S(j)$ distribution determines the MN-specific R and C parameters of a cohort of $N$ LIF models, which is driven by the current input $I(t)$ and predicts (C) the spike trains $sp_j^{sim}(t)$ of the $N$ virtual MNs constituting the complete MN pool. (D) The MN instantaneous discharge frequencies were computed from the simulated the spike trains $sp_j^{sim}(t)$ and were smoothed with a sixth-order polynomial [55]. One out of ten IDFs is displayed for clarity. The blue-to-red gradient for small-to-large MNs shows that the output behaviour of the reconstructed MN population is in agreement with the Onion-Skin scheme.

trace $\overline{F}(t)$ (green trace in Fig 13). The $eND_N$ was accurately predicted for the datasets $DTA_{35}$ and $HTA_{35}$, with $r^2 = 0.97$–$0.98$ and $nRMSE < 8\%$ (Table 3). The results obtained from the dataset $HGM_{30}$ returned $r^2 = 0.89$ and $nRMSE = 18.2\%$ (Table 3), accurately predicting the $eND_N$ for the positive ramp and first half of the plateau of force and then underestimating the true neural drive ($HGM_{30}$, Fig 13). The $eND_N$ predicted for the dataset $HTA_{50}$ returned results

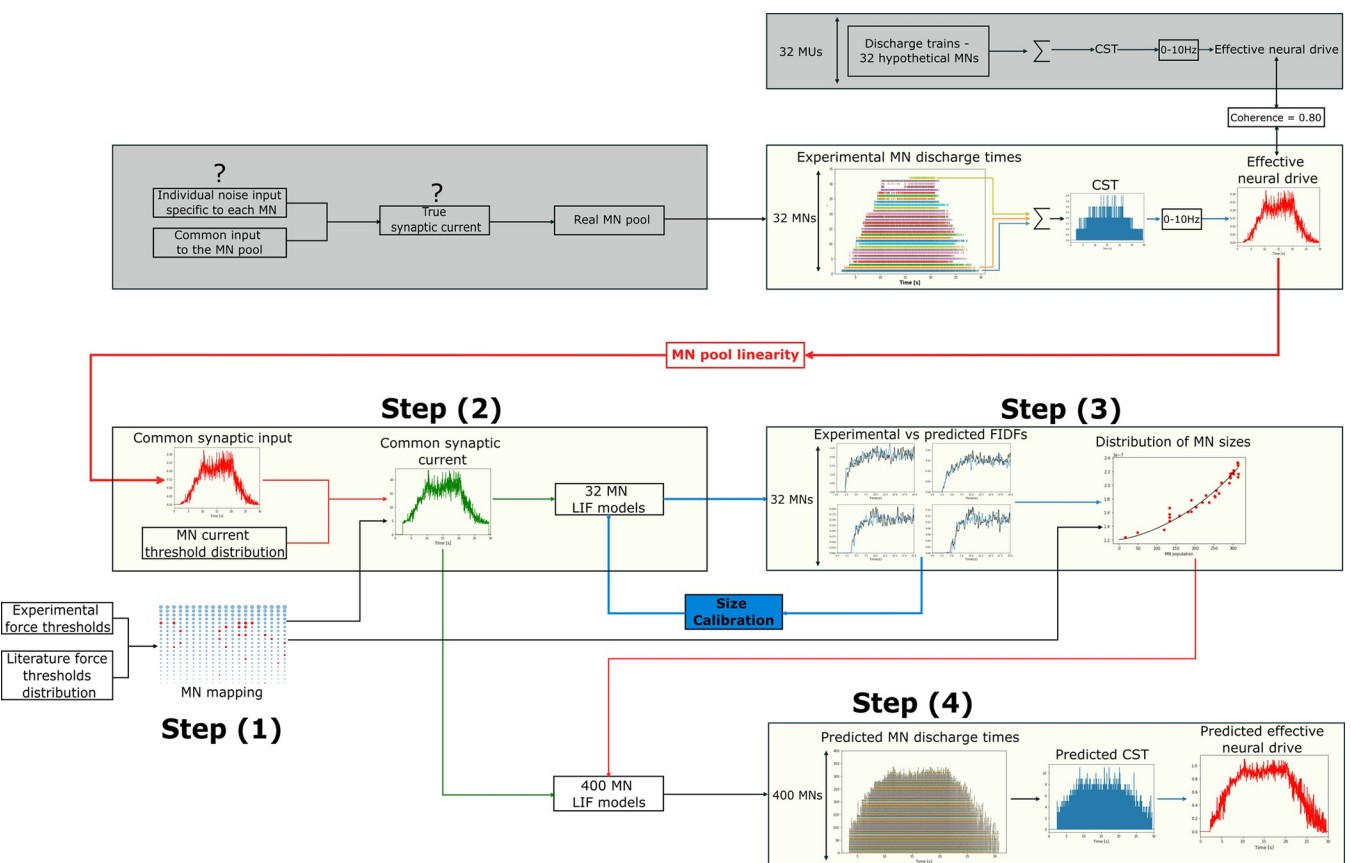

**Fig 11. Detailed description of the 4-step workflow applied to the $DTA_{35}$ dataset.** The firing activity of a fraction of the MN pool is obtained from decomposed HDEMG signals. These $N_r = 32$ experimental spike trains provide an estimate of the effective neural drive to muscle and explain most of the MN pool behaviour (*coherence* = 0.8). From a mapping of the $N_r$ identified MNs to the complete MN pool (Step (1)), literature knowledge on the typical distribution of $I^{th}$ in a mammalian MN pool, and using the linearity properties of the population of $N_r$ MNs, the current input $I(t)$ is estimated (Step (2)). $I(t)$ is input to $N_r$ LIF models of MN to derive, after a one-parameter calibration step minimizing the error between experimental and LIF-predicted FIDFs, the distribution of MN sizes across the complete MN pool (Step (3)). The distribution of MN sizes, which entirely describes the distribution of MN-specific electrophysiological parameters across the MN pool, scales a cohort of $N = 400$ LIF models which transforms $I(t)$ into the simulated spike trains of the $N$ MNs of the MN pool (Step (4)). The effective neural drive to muscle is estimated from the $N$ simulated spike trains.

of lower accuracy ($r^2$ = 0.88 and *nRMSE* = 15.1%) compared to the three other datasets. A null *eND* was predicted for one third of the simulation where a muscle force up to 20%*MVC* is generated, and the *eND* was overestimated for the rest of the (de)recruitment phase ($HTA_{50}$, orange trace, Fig 12). As detailed in the discussion, the latter is explained by an inadequate *IP* (*j*) distribution (Table 2), due to a lack of experimental information in the dataset $HTA_{50}$, which returns non-physiological maximum firing rates for the low-thresholds MNs. Two data points (16; 0.032) and (118; 0.037), obtained from the experimental data in the dataset $DTA_{35}$ with a scaling factor of $\frac{35}{50}$ applied to the *IP* values, were appended to the list of $N_r$ data points $(N_i; IP_{N_i})$ to describe the maximum firing behaviour of the first half of the MN pool for which no MN was identified for the dataset $HTA_{50}$ (Table 2). A new *IP*(*j*) distribution was obtained, returning an improved estimation of the $eND_N$ ($HTA_{50}$, purple trace, Fig 12) with respectively lower and higher *nRMSE* and $r^2$ metrics (Table 3). With three times lower *nRMSE* and higher $r^2$ values for all four datasets (Table 3), the $eND_N$ predicted from the 4-step approach (orange dotted traces in Fig 13) was a more accurate representation of the real effective neural drive than the $eND_{N_r}$ (blue dotted traces in Fig 13) computed from the $N_r$ experimental spike trains,

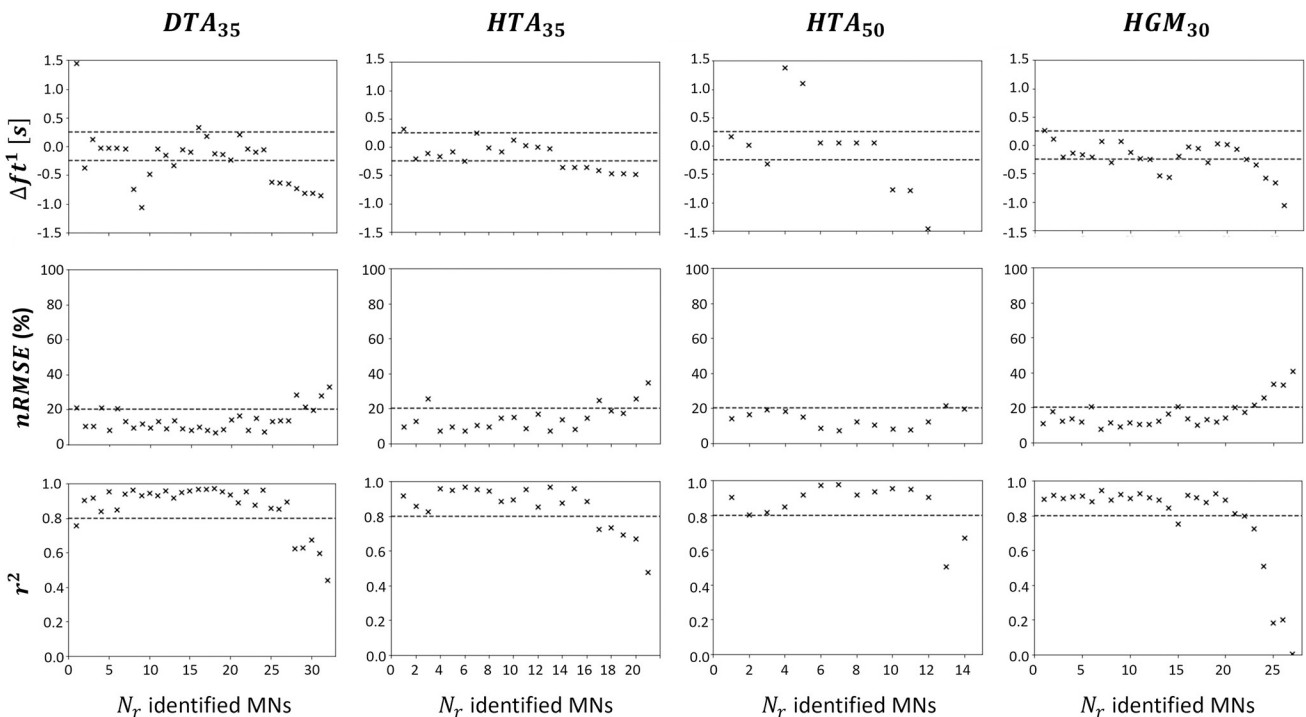

**Fig 12. Validation of the MN recruitment and firing behaviour predicted with the 4-step approach (Fig 1) for the $N_r$ MNs experimentally identified in the four datasets $DTA_{35}$, $HTA_{35}$, $HTA_{50}$ and $HGM_{30}$ described in Table 1.** For the validation of each $i^{th}$ predicted MN spike train, the experimental information of the $i^{th}$ identified MN was omitted when deriving the current input $I(t)$ and the $IP(j)$ and $S(j)$ distributions (steps 2 and 3), from which the $IP_i$ and $S_i$ parameters are obtained without bias for the $i^{th}$ MN. The spike train $sp_i^{sim}(t)$ of the $i^{th}$ MN is then predicted with an $IP_i$, $S_i$-scaled LIF model receiving $I(t)$ as input. The absolute error in predicting the experimental recruitment time $\Delta ft_i^1$ $(s)$ is reported for each of the $N_r$ MNs in the first row of the figure. Experimental and filtered instantaneous discharge frequencies $FIDF_i^{exp}(t)$ and $FIDF_i^{sim}(t)$, computed from $sp_i^{exp}(t)$ and $sp_i^{sim}(t)$ are compared with calculation of $nRMSE$ (%) and $r^2$ in the second and third rows of the figure. The dashed lines represent the $\Delta ft^1 \in [-250; 250]ms$, $nRMSE \in [0; 15]\%MVC$ and $r^2 \in [0.8, 1.0]$ intervals of interest respectively.

especially in the phases of MN (de)recruitment where the real effective neural drive was underestimated by $eND_{N_r}$. The predicted $eND_N$ also produced less noise than the $eND_{N_r}$ trace during the plateau of force. With $r^2 > 0.85$ and $nRMSE < 20\%$, this four-step approach is valid for accurately reconstructing the $eND$ to a muscle produced by a collection of $N$ simulated MN spike trains, which were predicted from a sample of $N_r$ experimental spike trains. It is worth noting that the accuracy of the $eND_N$ predictions, performed in Table 3 and Fig 13 for $N_{TA} = 400$ and $N_{GM} = 550$ MNs as discussed in the Methods Section, was not sensitive to the size $N$ of the reconstructed MN population. For example, for the $DTA_{35}$ dataset, the $eND_N$ computed with populations of $N = \{32, 100, 200, 300, 400\}$ MNs systematically compared to the real effective neural drive with $r^2 = 0.98$, and respectively returned $nRMSE = 7.7, 6.9, 6.5, 6.1$ and $5.9\%$, with slightly more accurate predictions of $eND_N$ with an increasing $N$ at low force during derecruitment.

## Application to MN-driven muscle modelling

Fig 14 reports for the datasets $DTA_{35}$ and $HTA_{35}$, for which the $eND_N$ was the most accurately predicted among the four datasets (Fig 13, Table 3), the time evolutions of the whole muscle force $F(t)$ recorded experimentally (green curve) and the whole muscle forces predicted with the MN-driven muscle model described in Fig 3 using $N_r$ experimental ($F_{N_r}(t)$, blue curves) and $N$ simulated ($F_N(t)$, red curves) spike trains as inputs. The muscle force trace $F_N(t)$

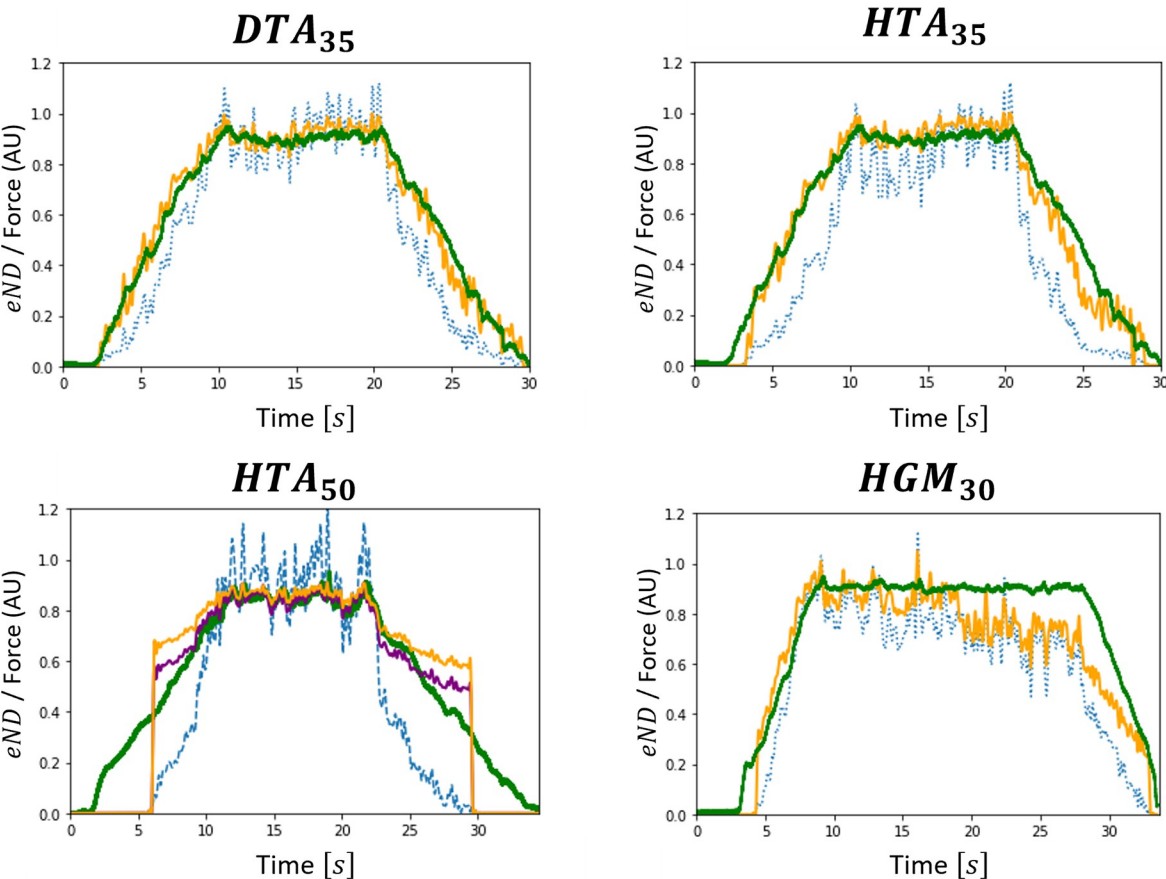

**Fig 13.** For the 4 datasets, validation against the normalized force trace $\overline{F}(t)$ (green trace) of the normalized effective neural drive to muscle ($eND_N$) produced by the complete MN pool (orange trace) and computed from the N MN spike trains $sp_j^{sim}(t)$ predicted with the 4-step approach. For comparison, the $eND_{N_r}$ directly computed from the $N_r$ identified MNs is also reported (blue dashed trace). For the dataset $HTA_{50}$, an additional prediction was performed (purple trace), where the IP(j) distribution obtained from step (3) was updated to return physiological maximum firing rates for all N MNs.

obtained from the *N* simulated spike trains derived in steps (1-4) compared with the experimental force *F(t)* with $r^2 = 0.95$ and $nRMSE = 11\%$ for the dataset $DTA_{35}$ and $r^2 = 0.97$ and $nRMSE = 11\%$ for the dataset $HTA_{35}$. These results validate the approach provided in this study for predicting the whole muscle force as a collection of MU force contributions from the

**Table 3.** For all four datasets, the $r^2$ and $nRMSE$ values obtained for the comparison of the time-histories of the normalized experimental force trace against the effective neural drive (1) $eND_{N_r}$ and (2) $eND_N$ computed from the spike trains of (1) the $N_r$ identified MNs and (2) the *N* virtual MNs. The results for $HTA_{50}$ (1) were obtained with the standard approach, while those for $HTA_{50}$ (2) were obtained with a revisited IP(j) distribution (see text).

| Dataset | $N_r$ MNs (experimental) | | N MNs (simulated) | |
|---|---|---|---|---|
| | $r^2$ | $nRMSE$ (%) | $r^2$ | $nRMSE$ (%) |
| $DTA_{35}$ | 0.92 | 19.5 | 0.98 | 5.9 |
| $HTA_{35}$ | 0.85 | 26.8 | 0.97 | 7.9 |
| $HTA_{50}$ (1) | 0.83 | 26.9 | 0.88 | 15.1 |
| $HTA_{50}$ (2) | | | 0.92 | 10.6 |
| $HGM_{30}$ | 0.85 | 28.8 | 0.89 | 18.2 |

MN-specific contributions to the effective neural drive. The muscle force trace $F_N(t)$ obtained from the $N$ MN spike trains returned more accurate predictions than $F_{N_r}(t)$ ($r^2 = 0.88$ and $nRMSE = 27\%$ for the dataset $DTA_{35}$ and $r^2 = 0.79$ and $nRMSE = 33\%$ for the dataset $DTA_{35}$), in which case the reconstruction of the MN pool in steps (1-4) was disregarded and the $N_r$ experimental spike trains were directly used as inputs to the muscle model in Fig 3.

## Discussion

This study reports a novel four-step approach, summarized in Fig 1 and displayed in detail in Fig 11, to reconstruct the recruitment and firing behaviour of a complete human pool of $N$ MNs from a sample of $N_r$ experimental spike trains obtained from the decomposition of HDEMG or intramuscular recordings during voluntary contractions. This approach can help neuroscientists, experimentalists, and modelers to investigate MN pool neuromechanics, better understand experimental datasets, and control more detailed neuromuscular models to advance our understanding of the neural strategies taken by the human central nervous system to control voluntary muscle contractions.

The three first steps of our approach identify from a sample of $N_r$ experimental spike trains a distribution of the MN electrophysiological properties across the MN pool. The $N_r$ MN spike trains are used to approximate the common synaptic input $CSI(t)$ to the complete MN pool (Fig 7). For simplicity, the $CSI$ is linearly related to the post-synaptic total dendritic membrane current $I(t)$, which is input to a cohort of $N_r$ single-compartment LIF models with current synapses. The LIF firing behaviour is entirely described by the phenomenological MN Inert Period parameter $IP_i$, derived from the experimental data (Fig 8), and by the MN size $S_i$ parameter to which all the MN electrophysiological properties are related, according to the relations provided in [42]. After calibration of the $S_i$ parameter, the $N_r$ LIF models accurately mimic the filtered discharge behaviour and accurately predict the recruitment dynamics of the $N_r$ experimental MNs (Fig 9). The $N_r$ MNs are allocated into the complete MN pool (Fig 5B and 5C) according to their recorded force recruitment thresholds $F_i^{th}$ (Fig 4B) and a typical species- and muscle-specific $F^{th}(j)$ distribution (Fig 5). From the previous findings, the continuous distribution of MN sizes $S(j)$ (Fig 10A) is derived for the complete pool of $N$ MNs. $S(j)$ defines the electrophysiological properties [42] of the MNs constituting the complete MN pool. The neural behaviour of the complete pool of $N$ MNs is predicted in the 4th step with a cohort of $N$ $S_j$-$IP_j$-scaled LIF models and the application of the current drive $I(t)$ (Fig 10).

### Validation of the approach

The approach was successfully validated both for individual MNs and for the complete pool of $N$ MNs. By blindly scaling the LIF models in steps (1-3) with permutations of $N_r-1$ input experimental spike trains, the filtered discharge behaviour and the recruitment dynamics of the $N_r$ individual MNs were accurately predicted for the four investigated datasets (Fig 12). The effective neural drive ($eND_N$) to muscle elicited by the complete pool of $N$ MNs was also accurately predicted by the 4-step approach for the $HTA_{35}$, $DTA_{35}$ and $HGM_{30}$ datasets (Fig 13, Table 3). The latter result suggests that the recruitment dynamics and the normalized distribution of firing rates across the firing MN pool were accurately predicted for the non-identified population of $N-N_r$ MNs. The accuracy of the $eND_N$ predictions was not sensitive to the size $N$ of the reconstructed population, with acceptable $eND_N$ predictions ($nRMSE < 20\%$, $r^2 > 0.9$) obtained with as few as ten distributed MNs. This suggests that the MN mapping (Step 1) and the derivation of the MN properties distribution (Step 3) in Figs 8 and 10A are the key contributions for describing the complete MN pool behaviour from the input experimental data.

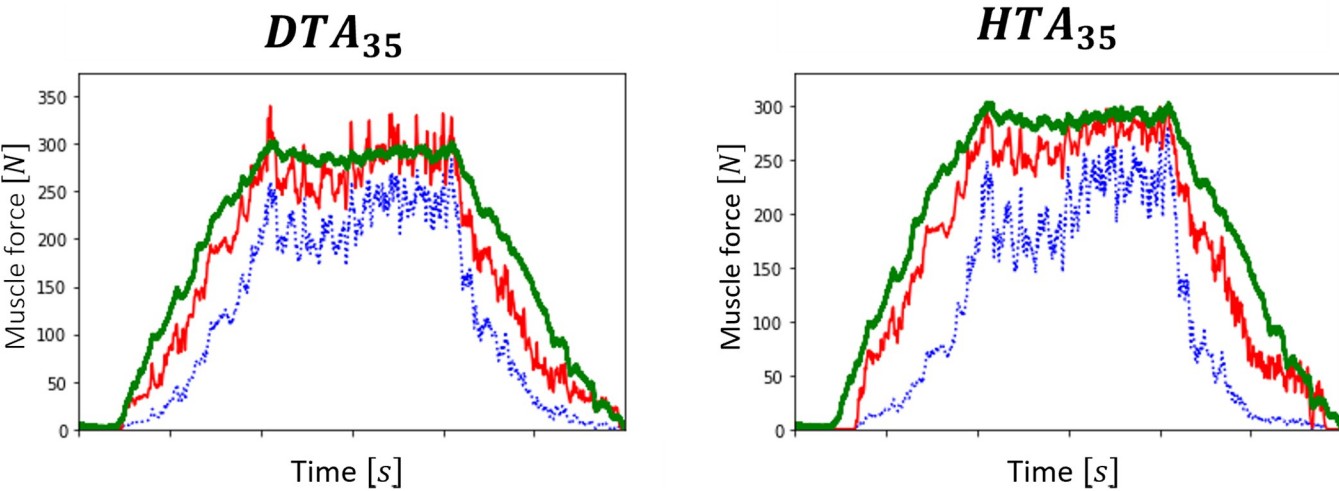

**Fig 14. Prediction of the experimental force trace $F(t)$ (green curve) with the MN-driven muscle model described in** Fig 3**.** $F_{N_r}(t)$ (blue curve) was predicted with the $N_r$ experimental spike trains input to the muscle model. $F_N(t)$ (red trace) was obtained from the cohort of $N$ experimental spike trains simulated by the four-step approach.

Despite the modelling limitations, discussed in [44] and in the Limitations Section, inherent to single-compartment LIF models with current synapses and related to the phenomenological $IP$, $C^d$ and $R^d$ parameters, the LIF models scaled following the methodology described in Step (3) can capture the distribution of MN discharging characteristics across the MN population. For example, the firing activity of the reconstructed MN population agrees with the onion-skin scheme (Fig 10D), as expected for slow low-force trajectories in human muscle contraction. While the MN size parameter calibration (Fig 9) only relies on the minimization of the root-mean square difference between experimental and predicted FIDFs, the normalized time-history of the FIDFs (high $r^2$ in Fig 12) and the MN recruitment times (low $\Delta ft^1$ values in Fig 12) of the $N_r$ MNs are both accurately mimicked (Fig 9) and blindly predicted (Fig 12) for all four datasets. Some realistic aspects of the MN neurophysiology are also maintained. For example, the range of MN surface areas $[0.15; 0.36] mm^2$ and input resistances $[0.5; 4.1] M\Omega$ obtained from the $S(j)$ distribution falls into the classic range of MN properties for cats [21,42], which is consistent with the distribution of rheobase values used in Step (2) that was obtained from cat data. The distribution of MN input resistance in the MN pool, obtained from the calibrated sizes $S(j)$, defines the individual MN rheobase thresholds and predicts that 80% of the human TA MU pool is recruited at 35%MVC (Fig 10C), which is consistent with previous findings in the literature [20,64].

## Benefits of the approach

**Benefit 1 – Better understanding the MN pool firing behaviour inferred from decomposed HDEMG outputs.** The workflow provides a method to better understand datasets of $N_r$ experimental spike trains obtained from signal decomposition. HDEMG decomposition techniques commonly identify samples of few tens of MNs [27] at most, i.e. typically less than 10-25% of the MU pool. The mapping in Fig 5 and Fig 10A demonstrates that these $N_r$-sized samples of MNs currently are not representative of the complete MN population, as supported by the $[0.65; 0.80]$ $coher_{N_r}$ values in Table 2. EMG decomposition shows a bias towards under-sampling small low-threshold MUs and identifying large high-threshold MUs because of their larger electric signal amplitude detected by the HDEMG electrodes. Consequently, the $CST_{N_r}$

computed from the $N_r$ identified MNs is not an accurate representation of the true muscle neural drive (Fig 13, Table 3). The method in Fig 11 identifies credible values for some MN-specific electrophysiological and morphometric properties, including MN membrane surface area, input resistance, capacitance, rheobase and time constant, for the dataset of $N_r$ experimental MNs and for the $N-N_r$ MNs that cannot be recorded. Using a realistic distribution of these properties, the $eND_N$ produced by the reconstructed MN population is accurately estimated, insensitive to the value of $N$, because it includes the firing activity of the complete fraction of the smaller MNs that were underrepresented in the experimental dataset, especially during the force ramps in $[t_1; t_2]$ and $[t_3; t_4]$(Fig 2), where the $eND_{N_r}$ can reach 70% normalized maximum error. The $eND_N$ besides includes the activity of the complete population of high-threshold MNs, which reduces noise (orange vs blue trace in Fig 13) during the plateau of force in $[t_2; t_3]$. The $eND_N$ also filters the local non-physiological variations displayed by the $eND_{N_r}$ such as the decrease in $eND_{N_r}$ in $[10; 15]s$ in $DTA_{35}$ and $HTA_{35}$ (top plots in Fig 13), the sudden rise and drop in $eND_{N_r}$ at $t = 10s$ and $t = 23s$ in $HTA_{50}$ (bottom left plot in Fig 13) or the steeply decreasing $eND_{N_r}$ from $t = 28s$ in $HGM_{30}$ (bottom right plot in Fig 13). Understanding and accounting for these experimental limitations is important and has critical implications in HDEMG-driven neuromuscular modelling when the $CST_{N_r}$ is input to a muscle model [109,110].

The workflow also provides a method to support future experimental investigations. The signal decomposition process can be refined and more MNs can be identified by iteratively running this workflow and using the $N-N_r$ simulated spike trains of the non-identified MNs as a model for identifying more MNs from the recorded signals. The method moreover provides simple phenomenological two-parameter-scaled LIF models described in step (3) to accurately replicate the recruitment and firing behaviour of the $N_r$ identified MNs (Fig 9) for further investigation.

**Benefit 2 – Advances in MN pool modelling.** This four-step approach advances the state-of-the-art in MN pool modelling. As discussed in the introduction, using a sample of $N_r$ experimental MN spike trains permits for the first time in human MN pool modelling to (1) estimate the true time-course of the common synaptic input to the MN pool, which cannot be measured experimentally, (2) approximate with a one-parameter calibration, available HDEMG data and knowledge from the literature, the MN-specific electrophysiological properties of all the MNs in the MN pool and a realistic distribution of these properties across the pool, which could, to date, only be obtained in specialized experimental and MN pool modelling studies in animals [39], and (3) validate the forward predictions of MN spike trains and effective neural drive to muscle to human experimental data. The pool of LIF models, the maximum firing frequency of which is obtained from the available experimental data, intrinsically accounts for the onion-skin theory [108] (Fig 10D), and better replicates the MN membrane dynamics of the MN pool than Fuglevand-type phenomenological models [10,29], where the MN-specific firing rates are predicted as a linear function of the current input $I(t)$. Moreover, single-compartment current-input LIF models can credibly replicate most of the MN membrane dynamics [43] and allowed accurate predictions of the MN pool behaviour (Figs 12 and 13), while they provide a simpler modelling approach and a more convenient framework for MN electrophysiological parameter assignment in the MN pool than comprehensive Hodgkin-Huxley-type MN models [30,32,111,112]. This four-step approach demonstrated to be robust to systematic differences in the input experimental datasets. For example, it accurately predicted the individual MN firing behaviour of the $N_r$ MNs for all four datasets (Fig 12) despite the latter being obtained at different levels of muscle contractions, on different subjects and muscles and in different experimental approaches. The approach accurately predicted the

$eND_N$ of both $DTA_{35}$ and $HTA_{35}$ datasets from $N_r$ = 32 and 21 identified MNs respectively (Table 3), suggesting that the method is not sensitive to the number $N_r$ of identified MNs, providing that the $N_r$ identified MNs and their properties are homogeneously spread in the MN pool, as in datasets $DTA_{35}$ and $HTA_{35}$ where at least one MN is identified in the each 10%-range of the rheobase domain. The accurate prediction of $eND_N$ for the dataset $HGM_{30}$ in the time range [4.7; 7.2] $s$ (Fig 13 bottom-right plot) also suggests that the quality of the predictions is not sensitive to the hindlimb muscle investigated, providing that the $F^{th}$ distribution is representative of the investigated muscle.

**Benefit 3 – Relevance for neuromuscular modelling.** A MN-driven neuromuscular model of in-parallel Hill-type actuators describing the MUs (Fig 3) is described in the Methods section and is used to predict the experimental muscle isometric force (green trace in Fig 14) from the $N_r$ experimental or $N$ reconstructed MN spike trains. The complete reconstruction of the discharging MN population detailed in Fig 11 is a key step towards advancing the state-of-the-art in neuromuscular modelling on several aspects.

Firstly, the neural input to the neuromuscular model in Fig 3 is a vector of experimental spike trains that is a more easily interpretable and a more detailed description of the muscle neural drive than the rectified-normalized-filtered envelopes of recorded bipolar EMGs (bEMGs) [95,113,114] or the $CST_{N_r}$ [109,110,115] typically used in single-input muscles models. Secondly, despite advanced multi-scale approaches in modelling whole muscles as a single equivalent MUs [98,116–120], the multiple Hill-MU model in Fig 3 provides a more convenient framework to model in detail the continuous distribution of the MU excitation-contraction and force properties in the MU pool. Thirdly, a few other multi-MU models were proposed in the literature [9,10,12,39,98,121,122], some of which used the Fuglevand's formalism [29] to model the MN firing behaviour and recruitment dynamics of complete pools of $N$ MNs, which are intrinsic to the experimental $sp_i^{exp}(t)$, to predict whole muscle forces with collections of $N$ MU Hill-type [9,123] Hill-Huxley-type [10] or twitch-type [39] muscle models. However, these studies lacked experimental neural control at the MN level and considered artificial synaptic inputs to their model, the predicted force of which was indirectly validated against results from other models and not against synchronously recorded experimental data, as performed in Figs 12–14 in this study. Finally, Fig 14 demonstrates that the reconstruction of the complete MN population described in this study (steps 1 to 4 in Fig 1) is a key step for accurate MN-driven neuromuscular predictions of muscle force. The $N_r$-MU model, that receives the $N_r$ individual MN spike trains $sp_i^{exp}(t)$, intrinsically underestimates the whole muscle activity when dominantly low-threshold MUs are recruited but are under-represented in the experimental sample (Figs 5 and 13). The $N$-MU model, which receives as inputs the $N$ spike trains $sp_j^{sim}(t)$ generated by the four-step approach, allows a more realistic assignment of distributed MU properties (MU type and maximum isometric force) to the complete MU population, and returned more accurate force predictions than the $N_r$-MU model (Fig 14). It is worth noting that the $N$-model did not require any parameter calibration except the MN size in step 3. The detailed modelling of the distribution of the excitation-contraction properties of the MUs makes the $N$-model more suitable for investigating the muscle neuromechanics than typical EMG-driven models, the neural parameters of which do not have a direct physiological correspondence and must be calibrated to match experimental joint torques [95,123,124].

## Limitations of the approach and potential improvements

Despite the aforementioned good performance of the four-step workflow, the method presents 4 main limitations, for some of which potential improvements are proposed in the following.

**Limitation 1 – Model validation.** The two validations of the approach (Figs 12 and 13) present some limitations. The local validation in Fig 12 only ensures that the method accurately estimates the MN firing behaviour for the fractions of the MN pool that were experimentally identified. This local validation alone does not inform on the accuracy of the predictions for the non-identified regions of the MN pool and must be coupled with a global validation of the MN pool behaviour by validating the predicted $eND_N$. While the local validation was successful for the $HTA_{50}$ dataset (Fig 12), where less than 30% of the active MN pool is represented, it is inferred in Fig 13C that the individual MN firing rates were overestimated for the first half of the non-recorded MN pool. This local validation would be self-sufficient for experimental samples that contain a large and homogeneously spread population of identified MNs, obtained from decomposed fine-wire intramuscular electrode and HDEMG grid signals. The validation of the $eND_N$ is performed in this study against an experimental force recorded by a transducer (green traces in Fig 13), which accounts for the force-generating activity of both the muscle of interest and the synergistic and antagonistic muscles crossing the ankle. The experimental force trace measured in the TA datasets may be an acceptable validation metric as the TA muscle is expected to explain most of the recorded ankle torque in dorsiflexion. The TA muscle is indeed the main dorsiflexor muscle with a muscle volume and maximum isometric force larger than the flexor hallucis longus muscle, which moment arm is besides not aligned with the dorsiflexion direction and mainly acts for ankle inversion. However, the gastrocnemius lateralis, soleus, and peroneus muscles acts synergistically with the GM muscle for ankle plantarflexion and the experimental torque recorded in dataset $HGM_{30}$ may not be representative of the GM muscle force-generation activity and may not be a suitable validation metric for $eND_N$. In Fig 13, the decreasing $eND_N$ and $eND_{N_r}$ (orange and blue dotted traces) may accurately describe a gradually increasing sharing of the ankle torque between synergistic muscles initially taken by the GM muscle, which the constant validation trace does not capture. To answer such limitations, the predicted $eND_N$ should be validated against a direct measure of muscle force, which can be performed as in other recent studies [13,14,116] with ultrasound measurements of the muscle fascicle or of the muscle tendon concurrently obtained with (HD)EMG recordings of the muscle activity. Finally, these two validations do not provide any indication whether the parameters calibrated with a trapezoidal force profile would generalize, for the same subject, to another force trajectory or a trapezoidal force profile up to another force level and provide accurate predictions of the $N_r$ experimental spike trains and of the $eND_N$. To perform such validation, the parameter identification in Step 3 (Fig 1) would be overlooked. The $N_r^{new}$ spike trains identified from the HDEMG signals concurrently measured with the new force profile would be used to derive the new current input $I^{new}(t)$ (Step 2) to drive a cohort of $N$ MN models (Step 4), the characteristics of which would be defined by the $IP$ and $S$ distributions (Step 3) derived with the first contraction profile, to predict the new effective neural drive $eND_N^{new}$. The MN mapping (Step 1) would serve to identify the MNs in the reconstructed MN populations, the predicted MN spike trains of which should be compared against the $N_r$ identified $sp_i^{exp}(t)$. Performing this final validation step was however impossible in this study because there exists, to the authors' knowledge, no open-source datasets of edited HDEMG signals recorded for the same subject for different force profiles.

**Limitation 2 – Sensitivity of the method to input HDEMG data.** While the method predicts a list of simulated spike trains $sp_j^{sim}(t)$ and a $eND_N$ that more accurately describes the MN pool behaviour than the experimental $sp_i^{exp}(t)$ and $eND_{N_r}$, as discussed previously, the accuracy of these predictions (Figs 12 and 13) remains sensitive to the distribution in the entire MN pool of the $N_r$ MNs identified experimentally, reported in the third column of Table 2. Because

of the definition of the current input $I(t)$, the $eND_N$ onset is defined by the recruitment time $tf_{N_1}^1$ of the lowest-threshold MN $N_1$ identified experimentally, as shown in Fig 13, while the unknown firing behaviour for $t < ft_{N_1}^1$ of the non-identified MNs of rheobase lower than $I_{N_1}^{th}$ is not captured. This is not an issue in samples of homogeneously distributed MNs like dataset $DTA_{35}$, where the 9th smallest MN (first 2% of the pool of threshold-increasing MNs) is identified (Fig 5BC), Table 2) and the $eND$ obtained from the 15 first MNs is not predicted during the short time range $[2.1, 2.3]s$, where the whole muscle builds only 1%MVC (top-left plot in Fig 13). However in heterogeneous or incomplete samples like dataset $HTA_{50}$, the lowest-threshold (232nd, Table 2) identified MN identified is recruited at $t = tf_{N_1}^1 = 6.1s$ and the approach wrongly returns a zero muscle neural drive $eND = 0$ for the time period $[1.6; 6.1]s$ where the muscle actually builds 20%MVC during the ramp of contraction (bottom-left plot in Fig 13). To tackle this issue, the normalized force trace, which is non-zero for $t < ft_{N_1}^1$ and representative of the CSI in this time region, could be scaled and used in lieu of the current definition of $I(t)$. However, this approach, which may not be suitable for non-isometric conditions, is not applicable in forward predictions of unknown whole muscle force from neural inputs. Experimental samples of homogeneously distributed MNs are also required to derive realistic $S(j)$ and $IP(j)$ distributions with the four-step approach. As observed in Fig 12, ignoring in the derivation of $I(t)$, $IP(j)$ and $S(j)$ the spike trains $sp_i^{exp}(t)$ of the MNs that are representative of a large subset of the MN pool affects the accuracy of the predicted recruitment and firing behaviour of the MNs falling in that subset (Fig 12). More specifically, the non-physiological distribution $IP(j)[s] = 5.6 \cdot 10^{-4} \cdot j^{0.80}$ was fitted to the experimental data of the 14 high-threshold MNs identified in dataset $HTA_{50}$, where the neural information of the 60% smallest MNs of the MN pool (Table 2) is lacking. Such $IP(j)$ distribution overestimates the LIF-predicted maximum firing frequency of low-threshold MNs, which explains the overestimation of the $eND_N$ in Fig 13 (bottom-left plot, orange curve). Non-physiological predictions can be avoided by adding artificial data points consistent with other experiments or with the literature in the rheobase regions of the MN pool where no MNs were experimentally identified. For example, using an $IP(j)$ relationship consistent with a dataset of homogeneously distributed MNs ($DTA_{35}$) constrained the predicted maximum firing rates to physiological values and returned more accurate predictions of the $eND_N$ (bottom-left plot, purple curve).

**Limitation 3 – LIF MN modelling.** The LIF MN model described from (Eq 7) to (Eq 13) was shown, with credible sets of inter-related parameters $S_{MN}$, $R$, $C$ and $\tau$ after [42], to accurately mimic (Fig 9) and blindly predict (Fig 12) most of the key phenomenological features of the $N_r$ firing MNs, including their FIDFs and time of first discharge, as well as nonlinear behaviours such as firing rate saturation (Fig 8) and hysteresis (Fig 4B and 4C). However, this MN model is mostly phenomenological and does not provide a realistic description of the actual mechanisms underlying the dynamics of action potential firing for several reasons. First, this single-compartment approach neglects the activity of the MN dendrites, which account for more than 95% of the total MN surface area and gather most of the post-synaptic receptors and MN PIC-generating channels responsible for the MN nonlinear input-output functions [40]. The inherent difference in membrane voltage dynamics between the dendrites and the soma, also partially mediated by hundred-fold differences in the value of passive electrophysiological parameters such as $R$ which increases with somatofugal distance [125] is therefore neglected in this point model approach. Then, the nonlinear dendritic activity being overlooked, the Common Synaptic Input CSI(t) in (Eq 5), which is the common net excitatory synaptic influx, is non-physiologically assimilated with a constant gain to the post-synaptic total dendritic membrane current $I(t)$ that depolarizes the MN soma and is responsible for spike generation. With this linear $CSI–I$ scaling in (Eq 6), the approach neglects the realistic

description of the voltage-driven dynamics of the PIC-generating channels that are responsible for some important nonlinear mechanisms, such as firing rate saturation [41] and hysteresis in the MN's current-voltage relation [93,94], which dictate the nonlinear *CSI–I* transformation [40]. In this study, the firing rate saturation, which in real MNs is due to a decrease in driving force for synaptic current flow as the dendrites become more depolarized [41], is captured by the phenomenological *IP* parameter. The observed hysteresis between the recruitment and decruitment thresholds (Fig 4B) and in the current-firing rate relation between recruitment and derecruitment phases (Fig 4C), explained by an hysteresis in the MN's current-voltage relation that arises from the activity of long-lasting PICs [20], are successfully but non-physiologically addressed by a tuning during derecruitment of the *R* and $C_m$ parameters respectively, although the values of the passive property *R* and of $C_m$, biological constant for the MN population [126], should be independent from the MN activity [44]. For these reasons, even if the distributions of the $S_{MN}$, *R*, *C* and $\tau$ parameters at recruitment take values consistent with the literature, as discussed, the MN LIF model described in (Eq 7) to (Eq 12) is not suitable for investigating the neurophysiology and the neuromechanics of individual MNs. The phenomenological tuning of the *IP*, *R* and $C_m$ parameters, although symptomatic of underlying nonlinear mechanisms, does not provide any insight into the true MN neurophysiology. If a more realistic MN model was considered in lieu of the LIF model, such as multiple-compartments Hodgkin-Huxley-based approaches [127], the four-step workflow would be an adequate tool for testing neurophysiological hypotheses and investigating some mechanisms and properties of the complete MN pool that cannot be directly observed experimentally in conditions of human voluntary contractions. A possible trade-off would be considering a single-compartment LIF model with active conductances [44], in which case (Eq 7) is re-written as (Eq 18), where $g_R = \frac{1}{R}$ is the constant MN passive conductance, $E_+ > \Delta V_{th}$ is the constant reversal potential for excitatory channels, $g(t, CSI)$ the time-varying total active excitatory synaptic conductance, and $I_{ps}(t, CSI, V_m) = g(t, CSI) \cdot (E_+ - V_m)$ the post-synaptic total dendritic membrane current induced by the synaptic drive *CSI*, the driving force of which is decreased by the term $(E_+ - V_m)$ when the dendritic membrane depolarizes, as discussed.

$$\frac{dV_m}{dt} = \frac{1}{C} \cdot \left[ g(t, CSI) \cdot (E_+ - V_m) - g_R \cdot V_m \right] \tag{Eq 18}$$

In the literature, $g(t, CSI) = g_{max} \cdot T(t, CSI)$ where $g_{max}$ is the maximum active conductance of the synapse and $T(t, CSI)$ can be interpreted as the fraction of bound synaptic receptors [30] or of opened ionic channels in the range [0; 1]or as a train of synaptic pulses [44]. Because setting $T(t, CSI) = CSI(t)$, with $CSI(t)$ as defined in Fig 6B, does not meet the requirements in firing rate saturation for large CSI input, the saturation in dendritic ionic channel activation for large CSI [41] must be accounted for and $T(t, CSI)$ should be a nonlinear saturating function of CSI. In this approach, the MN-specific hysteresis and firing rate behaviour could be obtained with a tuning during recruitment and derecruitment and a distribution across the MN pool of the $g_{max}$ parameter and of the shape parameter of the *S* function based on the experimental data. Although this approach more realistically describes the neurophysiological mechanisms underlying the nonlinear MN behaviour during discharge events, it is more challenging to scale using the experimental information. Considering that this study focuses on the phenomenological behaviour of individual MNs and on the overall activity of reconstructed MN populations, the LIF model defined in (Eq 7) was judged an adequate trade-off between accuracy and modelling complexity, considering the overall accurate predictions of the MN firing behaviours (Fig 12) and its low computational cost, and no other modelling approaches such as (Eq 18) were pursued.

It must be noted that, whatever the MN modelling approach, the experimental drive to the MN currently considered in the four-step method is the CSI, which disregards the MN-specific synaptic noise $SN(t)$, which is responsible for most of the inter-spike variability (ISV). Considering that the MN pool and the MU neuromechanical mechanisms are expected to filter the MN-specific $SN(t)$, this simplification is adequate for accurate predictions of the $eND_N$ and of the whole muscle force [83]. Sometimes modelled as a random Gaussian-like event [29], the ISV can be obtained as the response to a $SN(t)$ white noise added to the current definition of the synaptic input as $CSI(t)+SN(t)$. The relative proportion of $CSI$ and $SN$ in % of the variance of the total synaptic current can be approximated from Fig 2A in [47]. Accounting for $SN(t)$ might improve the accuracy of the predicted firing behaviour of the largest MNs, for which the largest $\Delta ft^1$ and $nRMSE$ and lowest $r^2$ values were obtained in Fig 12 for all four datasets, and the firing behaviour and recruitment dynamics of which are dominantly dictated by random fluctuations of $SN(t)$.

**Limitation 4 – Limited experimental data in the literature.** The four-step approach is constrained by the limited knowledge in the literature of the characteristics of the human MU pool. Therefore, the LIF parameters and the MN rheobase distribution in the MN pool (Fig 7A) are tuned with typical cat data from various hindlimb muscles [42] while the experimental MN spike trains were obtained from the human TA muscle. While the normalized mathematical relationships relating the MN electrophysiological parameters of the LIF model ($R$, $C$, $\tau$, $\Delta V_{th}$) can be extrapolated between mammals [42] and thus to humans, no experimental data is yet available to scale these relationships to typical human data.

Also, the typical $F^{th}(j)$ and $I_{th}(j)$ threshold distributions, derived for mapping the $N_r$ identified MNs to the complete MN pool (Fig 5) and for scaling the $CSI$ to physiological values of $I(t)$ (Fig 7), were obtained from studies which relied on experimental samples of MN populations of small size. These source studies therefore did not ensure that the largest and lowest values were identified and reported, and that the identification process was not biased towards a specific subset of the MN pool, such as larger MNs. In such cases, the true threshold distributions would be more skewed and spread over a larger range of values, as discussed in [42], than the distributions reported in Figs 5 and 7. The $F^{th}(j)$ distribution is besides muscle-specific [20] with large hindlimb muscles being for example recruited over a larger range of MVC than hand muscles. However, enough data is reported in the literature to build the $F^{th}(j)$ distributions for the TA and first dorsal interossei human muscles only. For these limitations, the two first steps of this approach could be made subject- species- and muscle-specific by calibrating the $F^{th}(j)$ and $I^{th}(j)$ described as the 3-parameter power functions defined in this study.

**Conclusion.** This study presents a four-step workflow (Figs 1 and 11) which predicts the spiking activity of the discharging MNs that were not identified by decomposed HDEMG signals. The method is driven by the common synaptic input, which is derived from the experimental data, and reconstructs, after a calibration step, the distribution across the MN population of some MN properties involved into the MN-specific recruitment and spiking behaviour of the discharging MNs (Figs 8 and 10A). The method blindly predicts the discharge behaviour of the $N_r$ experimentally identified MNs (Fig 12) and accurately predicts the muscle neural drive (Fig 13) after the complete discharging MN population is reconstructed (Fig 10C and 10D). With direct application in neuromuscular modelling (Fig 14), this method addresses the bias of HDEMG identification towards high-threshold large units and is relevant for neuroscientists, modelers and experimenters to investigate the MN pool dynamics during force generation.

## Author Contributions

**Conceptualization:** Arnault H. Caillet, Andrew T. M. Phillips, Dario Farina, Luca Modenese.

**Formal analysis:** Arnault H. Caillet.

**Investigation:** Arnault H. Caillet.

**Methodology:** Arnault H. Caillet, Dario Farina, Luca Modenese.

**Resources:** Dario Farina.

**Software:** Arnault H. Caillet.

**Supervision:** Andrew T. M. Phillips, Dario Farina, Luca Modenese.

**Validation:** Arnault H. Caillet.

**Visualization:** Arnault H. Caillet.

**Writing – original draft:** Arnault H. Caillet.

**Writing – review & editing:** Arnault H. Caillet, Andrew T. M. Phillips, Dario Farina, Luca Modenese.

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
