## [Decision Letter · Decision Letter 0]

2 Jun 2022

Dear Dr. Modenese,

Thank you very much for submitting your manuscript "Estimation of the firing behaviour of a complete motoneuron pool by combining electromyography signal decomposition and realistic motoneuron modelling" for consideration at PLOS Computational Biology.

As with all papers reviewed by the journal, your manuscript was reviewed by members of the editorial board and by several independent reviewers. In light of the reviews (below this email), we would like to invite the resubmission of a significantly-revised version that takes into account the reviewers' comments.

We cannot make any decision about publication until we have seen the revised manuscript and your response to the reviewers' comments. Your revised manuscript is also likely to be sent to reviewers for further evaluation.

Sincerely,

Andrew J Fuglevand

Guest Editor

PLOS Computational Biology

Lyle Graham

Deputy Editor

PLOS Computational Biology

Reviewer's Responses to Questions

**Comments to the Authors:**

Reviewer #1: This study describes the use of a set of leaky integrate-and-fire (LIF)motoneuron models to predict the recruitment and firing behavior of the complete pool of motoneurons (MNs) innervating a muscle, based on experimental recordings of a subset of MN spike trains during a trapezoidal isometric contraction. The authors demonstrate that the parameters of their LIF models can be tuned to recreate the firing behavior of the recorded MNs, and can also be used to predict the firing behavior of the entire pool. Using the activation of MNs from the entire pool as an input to a muscle model gives a much better fit to the normalized torque recorded during the contractions than does the input from just the recorded spike trains. This approach can thus help experimentalists and modelers better understand the relation between force production and the MN drive to the muscle. However, the approach does not provide much insight into the biophysical mechanisms controlling MN firing rate behavior given the many differences between a single-compartment model with no active conductances and real motoneurons with a variety of active conductances and an extended dendritic tree.

Lines 96 - 97. For a consideration of MN-specific nonlinear mechanisms, you should also cite Binder et al. (Nonlinear Input-Output Functions of Motoneurons, Physiology,35(1):31-39,2021) and Powers and Heckman (Synaptic control of the shape of the motoneuron pool input-output function, J Neurophys, 117(3):1171-1184, 2017).

Line 282. The common synaptic current I(t) that drives spike generation is the total dendritic membrane current that arises not just from the net excitatory synaptic influx but also from intrinsic current generated by voltage-dependent Na and Ca channels.

Lines 322 – 368. In this section or in the Discussion you should note that the IP parameter characterizes firing rate saturation that in actual motoneurons is likely to be due to a decrease in driving force for synaptic current flow as the dendrites become more depolarized.

Lines 334 – 336. How is the IDF trendline calculated? Some kind of smoothing? A polynomial? It would be good to add plot to Figure 2 that shows an example of the IDF trendline superimposed on the instantaneous discharge frequencies.

Lines 361 – 367. As was the case for the IP parameter, the change in Rm to induce firing rate hysteresis does not reflect the actual mechanism, which likely arises from hysteresis in the motoneuron’s current-voltage relation (cf. Lee and Heckman, J Neurophys 1998ab).

Line 405. I’m not aware of any estimate of specific membrane capacitance as high as 10.8 microF/cm2. Please provide a reference.

Lines 414 – 426. Changing Cm during the course of a contraction may provide a good fit to the firing rate near derecruitment, but it doesn’t provide any insight into the underlying mechanisms governing derecruitment behavior. The only mechanism I can think of to alter the measured capacitance of an area of membrane on this time scale would be to change the area by endo- or exocytosis.

In Figure 4B it would be useful to show both the line of best fit and the line of identity.

In Figure 4C, the fit to the instantaneous discharge frequencies would be clearer if the IDFs were plotted as symbols rather than lines from zero.

Please comment on the discrepancy between the time course the low pass filtered CST and the recorded force in Figure 6B. Presumably this reflects under-sampling of low threshold units.

Figure 9A and B. Why is the ordinate scale for FIDFs from 0 to around 0.25? Shouldn’t it be 0 to 25? The authors should add panels that show how good the fits are as the unit is derecruited.

Lines 584 – 590. The surface area in the model is for a single isopotential compartment, whereas motoneurons have extended dendritic structures. The values for surface area in Table 2 and Figure 10 are lower than what is typically seen for total surface area in cat motoneurons which are about three times the values shown here.

Lines 693 – 703. The values of Cm that work best for a single compartment LIF model may have little to do with specific membrane capacitance values for real motoneurons. Variations in time constant and input resistance in real motoneurons are mostly strongly influenced by changes in specific membrane resistivity rather than capacitance or even size (see Gustafsson and Pinter, JPhysiol, 356: 401-431, 1984).

Lines 709 – 723. As mentioned above, changing Cm over the course of contraction doesn’t make a lot of sense, even if it does improve the fit for firing during decreasing drive. Figures 14 and 15 are not necessary and distract from the most important findings of the study presented in Figures 13 and 16, namely the improved prediction of force when the complete simulated MN pool is used to drive the muscle as opposed to just the recorded spike trains.

Lines 828 – 841. As mentioned above, the vast differences between a single compartment LIF model and real motoneurons preclude a simple correspondence between LIF parameter values (e.g., size and Cm) and values in actual motoneurons.

Limitations of the approach. The authors should include a section on the uncertain relation between LIF parameters and biophysical mechanisms.

Line 941. Once again, I cannot think of a mechanism for a time-history-dependent C parameter

Reviewer #2: My review is uploaded as an attachment.

**Have the authors made all data and (if applicable) computational code underlying the findings in their manuscript fully available?**

Reviewer #1: Yes

Reviewer #2: **No: **The authors state, "Computational code is available under request to the authors." This is not consistent with PLOS policies.

PLOS authors have the option to publish the peer review history of their article (what does this mean?). If published, this will include your full peer review and any attached files.

Reviewer #1: **Yes: **Randall Powers

Reviewer #2: **Yes: **Kelvin Jones
---

## [Decision Letter · Decision Letter 1]

2 Sep 2022

Dear Dr. Modenese,

Thank you very much for submitting your manuscript "Estimation of the firing behaviour of a complete motoneuron pool by combining electromyography signal decomposition and realistic motoneuron modelling" for consideration at PLOS Computational Biology. As with all papers reviewed by the journal, your manuscript was reviewed by members of the editorial board and by several independent reviewers. The reviewers appreciated the attention to an important topic. Based on the reviews, we are likely to accept this manuscript for publication, providing that you modify the manuscript according to the review recommendations.

Sincerely,

Andrew J Fuglevand

Guest Editor

PLOS Computational Biology

Lyle Graham

Section Editor

PLOS Computational Biology

[LINK]

Reviewer's Responses to Questions

**Comments to the Authors:**

Reviewer #1: The authors have done a very thorough job of addressing my concerns. As they point out in their reply, there is a trade-off between the advantages of their LIF models (specifically their simplicity and their prediction accuracy) and the fact that they provide less insight into underlying biophysical mechanisms than more complex Hodgkin-Huxley type models. The benefits and limitations of the LIF approach are now considered in more detail in the revised manuscript. The authors have also removed the use and analysis of a time-varying membrane capacitance which as both Kelvin and I pointed out is quite non-physiological. My only other suggestion for revision is that in section Limitation 3 in the discussion, the authors in might want to mention the use of a two-compartment model (soma and dendrite, e.g., Kim et al. Plos one 9, e95454, 2014) as a potential avenue for future research.

**Have the authors made all data and (if applicable) computational code underlying the findings in their manuscript fully available?**

Reviewer #1: Yes

PLOS authors have the option to publish the peer review history of their article (what does this mean?). If published, this will include your full peer review and any attached files.

Reviewer #1: **Yes: **Randall Powers

Figure Files:

Data Requirements:

Reproducibility:

References:

---

## [Editor Report · Decision Letter 2]

8 Sep 2022

Dear Dr. Modenese,

We are pleased to inform you that your manuscript 'Estimation of the firing behaviour of a complete motoneuron pool by combining electromyography signal decomposition and realistic motoneuron modelling' has been provisionally accepted for publication in PLOS Computational Biology.

Best regards,

Andrew J Fuglevand

Guest Editor

PLOS Computational Biology

Lyle Graham

Section Editor

PLOS Computational Biology

---

## [Editor Report · Acceptance letter]

22 Sep 2022

PCOMPBIOL-D-22-00412R2 

Estimation of the firing behaviour of a complete motoneuron pool by combining electromyography signal decomposition and realistic motoneuron modelling

Dear Dr Modenese,

I am pleased to inform you that your manuscript has been formally accepted for publication in PLOS Computational Biology. Your manuscript is now with our production department and you will be notified of the publication date in due course.

With kind regards,

Zsofia Freund
